# Region-specific defect engineering of $Bi_2W_{1-x}O_{6-\gamma}$ induces nanoscale electric fields and surface active-sites for enhanced visible-light oxidation of salt-lake flotation agents

Liang Ma[1,2], Siyuan Zhang [ID][1] ✉, Haining Liu[1], Chunyan Wang[1,2], Zhongmei Song[1,2], Wenjie Han[1], Mingzhe Dong[1], Jungang Hou[3] ✉, Weidong Shi[4] ✉ & Xiushen Ye [ID][1] ✉

Breaking the limitations of conventional defect engineering, this work pioneers region-specific dual-defect engineering in $Bi_2WO_6$. By precisely tailoring tungsten (W) and oxygen (O) vacancies at nanoscale spatial domains-W vacancies at the edges and O vacancies at the center-a spatially asymmetric defect configuration is achieved. This configuration induces a synergistic "defect dipole" effect, amplifying the internal electric field by 2.74 times while simultaneously enriching surface-active sites. As a result, the photocatalytic efficiency is dramatically enhanced, achieving complete oxidation of recalcitrant flotation agents-octadecylamine (ODA) and 4-dodecylmorpholine (DMP)-within just 2 h of visible light irradiation, which is 3.6 times faster than that of pristine $Bi_2WO_6$. Additionally, the generation of reactive species ($\cdot O_2^-$, $^1O_2$, and $h^+$) is significantly boosted by factors of 8.98, 5.55, and 20.02, respectively, highlighting the material's remarkable reactivity. Photoelectrochemical analyses reveal a remarkable 290% increase in charge separation efficiency. This enhancement is further supported by an improved $O_2$ adsorption capacity, which promotes the formation of reactive oxygen species involved in the degradation process. Impressively, the engineered $Bi_2W_{1-x}O_{6-\gamma}$ exhibits outstanding performance in real-world industrial wastewater treatment under solar irradiation, demonstrating its practical viability. Overall, this work establishes a new paradigm in photocatalysis by integrating precise nanoscale defect engineering with enhanced electrostatic modulation.

The escalating global population and the imperative of global food security have catalyzed a significant surge in potassium fertilizer demand, a crucial plant nutrient[1]. The industrial production of potash primarily relies on froth flotation method, involving the use of organic flotation agents like octadecylamine (ODA) and 4-dodecylmorpholine (DMP). The prevalent practice in global potash fertilizer production of discharging large amounts of organic flotation agents in waste potash tailings slurry directly into the environment presents considerable potential risks to surface water, ecosystems, groundwater, and terrestrial environments[2]. Given that these organic pollutants, inert, toxic and non-biodegradable, persist in natural open environments, solar-driven photocatalytic mineralization stands out as a promising

strategy, offering a sustainable and environmentally compatible solution[3]. Among the diverse array of photocatalytic materials, bismuth-based oxides, particularly BWO ($Bi_2WO_6$) have attracted substantial attention due to their layered Aurivillius structure, suitable band gap, chemical stability, non-toxic nature, and exceptional visible light responsiveness[4–6]. However, pristine $Bi_2WO_6$ suffers from insufficient charge separation and transfer (CST), limited surface active sites, and inadequate visible light utilization efficiency, which significantly curtails its photocatalytic efficiency and practical environmental applications[7].

Recent advancements in defect engineering have opened new avenues for enhancing the photocatalytic performance through modulating the electronic structure, enhancing CST, and improving the surface reactivity of photocatalysts[8,9]. Recognized as a key anionic defect, oxygen vacancies have demonstrated remarkable potential in photocatalytic applications, particularly in enhancing material performance metrics. Surface-engineered oxygen vacancies in Aurivillius-phase nanosheets facilitate broadband visible light harvesting and accelerate photoinduced charge carrier spatial separation[10–12]. Concurrently, abundant oxygen vacancies dynamically transform the surface of the photocatalyst by offering a rich array of reactive sites, thus intensifying photocatalytic activity[5,13,14]. In particular, the incorporation of oxygen vacancies in a semiconductor generates defect dipoles that establish an internal electric field (IEF), thereby facilitating enhanced CST processes[15,16]. Contrarily, the deliberate introduction of cationic vacancy results in notable enhancements in photocatalytic performance, particularly in terms of improved CST and reactive sites. Engineered Bi vacancies generate structural strain within the lattice, subsequently enhancing the efficiency of CST processes[17]. A strategically constructed W vacancy gradient layer generates an internal homojunction with a strong IEF, facilitating efficient photoelectron migration to the surface and abundant reactive sites[18]. Furthermore, engineered tungsten vacancies induce critical mid-gap electronic states, enabling broadband visible light harvesting and facilitating more efficient CST processes[19].

The incorporation of individual anionic or cationic defects enhances photocatalytic activity through both shared and defect-specific mechanisms. Inspired by this established research, it is a plausible hypothesis that maintaining the independent enhancement mechanisms of dual defects for the photocatalytic performance, alongside a deliberate amplification of their shared enhancement effects, could provide a robust framework for achieving significantly improved photocatalytic activity. Nevertheless, the potential synergistic effects of dual defects and their interactions with surface properties have not been comprehensively explored, attributed to the challenges in precisely controlling and characterizing dual defects, predicting and optimizing their combined effects, and maintaining structural stability[20]. How to design a facile, cost-efficient approach for strategic co-engineering of W and O vacancies in pristine $Bi_2WO_6$ remains important yet challenging. Naturally, a comprehensive examination is imperative to elucidate the multifaceted impacts of dual anionic-cationic defects on charge separation dynamics, light absorption characteristics, and surface catalytically active sites. Therefore, addressing these challenges is essential to unlock the full potential of dual-defect engineering, paving the way for new frontiers in photocatalytic material design.

Herein, a strategic one-step in situ dual-defect engineering approach is developed, enabling precise control over W and O vacancy formation in $Bi_2WO_6$ systems. Characterization reveals a 2.7-fold enhancement in IEF strength, significantly improving CST processes. The engineered electronic structure, featuring optimized band gap and enhanced light absorption properties, results from the introduction of intermediate energy states. Surface analysis confirms increased reactive site density, leading to enhanced generation of reactive oxygen species, particularly $^1O_2$ and $O_2^-$. Implementing precise defect

engineering, the $Bi_2WO_6$ material manifests exceptional degradation performance, dramatically increasing organic flotation reagent degradation rates by 3.6-fold and facilitating complete mineralization of target organic substances. The dual-defect $Bi_2WO_6$ photocatalyst demonstrates remarkable efficacy in rapidly degrading and completely mineralizing high-salinity wastewater from an actual potassium fertilizer industrial under authentic outdoor solar irradiation conditions, thus demonstrating compelling real-world application potential.

## Results

The synthesis of $Bi_2WO_6$ and its defect-engineered derivatives was primarily accomplished through the hydrothermal method (Fig. 1a). Controlled solvent selection facilitated the formation of pristine bismuth tungstate (BWO), W-deficient $Bi_2WO_6$ (BWO-S), O-deficient $Bi_2WO_6$ (BWO-E), and concurrent W-O deficiencies $Bi_2WO_6$ (BWO-ES) (Text S1). Ethylene glycol, as a reductive polyol, promotes bulk O vacancy generation by abstracting lattice oxygen and stabilizing defect structures. Meanwhile, NaOH, via high alkalinity and potential W leaching, largely localizes W vacancies to surface/edge domains. The defect concentration in synthesized materials can be precisely regulated through facile modulation of solvent composition during the reaction process, offering a straightforward approach to structural control (Supplementary Figs. 1 and 2, and Supplementary Table 1).

All observed X-ray diffraction (XRD) patterns were reliably indexed to the orthorhombic phase of $Bi_2WO_6$ (JCPDS No. 98-002-3584) (Fig. 1b). The introduction of O, W, and O-W co-defects did not significantly alter the characteristic diffraction peaks, indicating preserved crystal structure integrity. Notable, however, was the sequential diminution of peak intensities correlating with the progressive incorporation of defects, indicating a poorer crystallinity. As evidenced by the Raman spectra (Fig. 1c), the presence of two well-defined peaks at 797 and $825\,cm^{-1}$ is indicative of antisymmetric and symmetric Ag stretching modes within terminal O-W-O groups[18]. Enhanced O-W-O stretching vibrations were observed in the BWO-S sample, stemming from W atomic deficiencies. In addition, oxygen vacancy ($O_V$) introduction in BWO-E and BWO-ES samples systematically diminishes peak intensities of O-W-O stretching vibrations, primarily attributing to crystal structure deterioration and deformation[21]. The Fourier-transform infrared (FT-IR) analysis shows absorption bands at $570\,cm^{-1}$, $728\,cm^{-1}$ and $1070\,cm^{-1}$, which are characteristic of Bi-O vibration, tungstate chain vibration and W-O-W vibrations, respectively[5] (Fig. 1d).

The scanning transmission electron microscopy (SEM) results indicate a progressive decrease in particle size with the introduction of defects, accompanied by an increase in particle dispersion (Supplementary Fig. 3). This transformation naturally induces a change in mesoporous characteristics, evolving from the Type IV H3 hysteresis loop observed in BWO and BWO-S to the Type IV H2 hysteresis loop in BWO-E and BWO-ES. The specific surface area consistently increases, reaching its peak at $41.6\,m^2/g$ for BWO-ES (Supplementary Fig. 4). Transmission electron microscopy (TEM) results reveal the morphology transformation of BWO and BWO-S from initial rectangular or square nanosheets to circular or elliptical nanosheets of BWO-E and BWO-ES (Supplementary Fig. 5 and Fig. 2a). High-resolution TEM (HRTEM) imaging of BWO-ES samples provides clear lattice fringes, confirming that the synthesized samples possess a well-developed crystalline structure (Fig. 2b and Supplementary Fig. 6). Further energy dispersive spectroscopy (EDS) shows that the spatial distribution of Bi atoms is essentially uniform, while O atoms are more sparsely distributed in the central region but exhibit a more even spread at the edges (Fig. 2c). In contrast, W atoms display a greater concentration at the center and a reduced presence at the edges (Fig. 2c). EDS analysis of BWO, BWO-S, and BWO-E further substantiates these observations (Supplementary Fig. 7), confirming that the dual-defect strategy has successfully induced W and O defects,

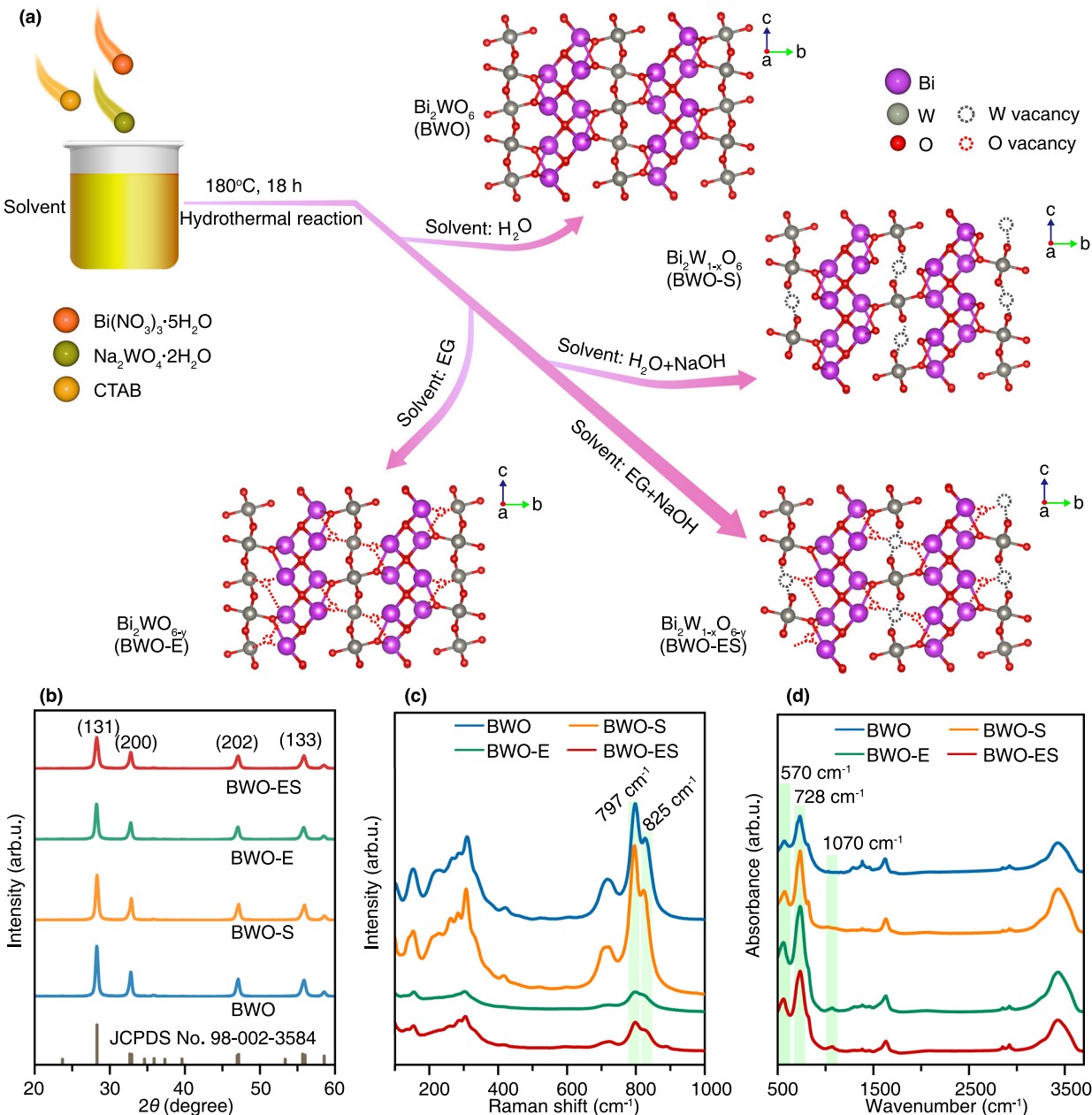

**Fig. 1 | Synthesis and characterizations. a** the schematic synthesis processes, **b** XRD patterns, **c** Raman spectra, and **d** FT-IR spectra of BWO, BWO-S, BWO-E and BWO-ES.

with W defects predominantly located at the edges. HRTEM EDS elemental line scanning was employed to further elucidate the spatial distribution of elements, with results shown in Fig. 2d, e. The pristine BWO sample demonstrated homogeneous distribution of Bi, W, and O across the entire scanning range. However, the BWO-ES sample exhibited markedly different distribution patterns, characterized by constant Bi levels, a gradual increase in O molar ratio from center to periphery, and a corresponding decrease in W molar ratio (Supplementary Fig. 8). The extent of atomic deficiencies was quantitatively assessed X-ray fluorescence (XRF) spectroscopy, revealing that dual-defect BWO-ES sample exhibited the most pronounced reduction in W and O content relative to BWO, BWO-S, BWO-E samples (Fig. 2f). These results were further validated through systematic EDS analysis (Supplementary Fig. 9).

To elucidate the spatial distribution of W and O vacancies in BWO-ES, disordered regions were systematically examined using high-angle annular dark-field SEM (HAADF-STEM) in conjunction with EDS

analysis. Examination of the HAADF intensity gradient along the white arrow in Fig. 2g reveals a systematic decrease (Fig. 2h), with concurrent Bi and W atomic overlap along [001] orientation (Fig. 2g). Despite the challenges in direct vacancy identification due to atomic overlap, quantitative analysis of Bi atomic ratio constant (Fig. 2e) conclusively establishes the presence of edge-localized W vacancies. Further evidence for the absence of W and O in BWO-ES is provided by STEM paired with electron energy loss spectroscopy (EELS) analysis in Fig. 2i, j. In the W M4-edge STEM-EELS spectra, a decrease in peak intensity and a shift to higher energy values with a gradual progression from the central to the edge are observed, which implies the absence of W atoms at the edge of BWO-ES[18,22]. Conversely, the oxygen K-edge STEM-EELS spectra show a gradual increase in peak intensity and a shift to lower energies from the center to the edge, suggesting that oxygen vacancies are primarily concentrated in the central region[23]. Furthermore, atomic force microscopy (AFM) characterization of the nanosheet-like BWO-ES samples showed that their thickness was

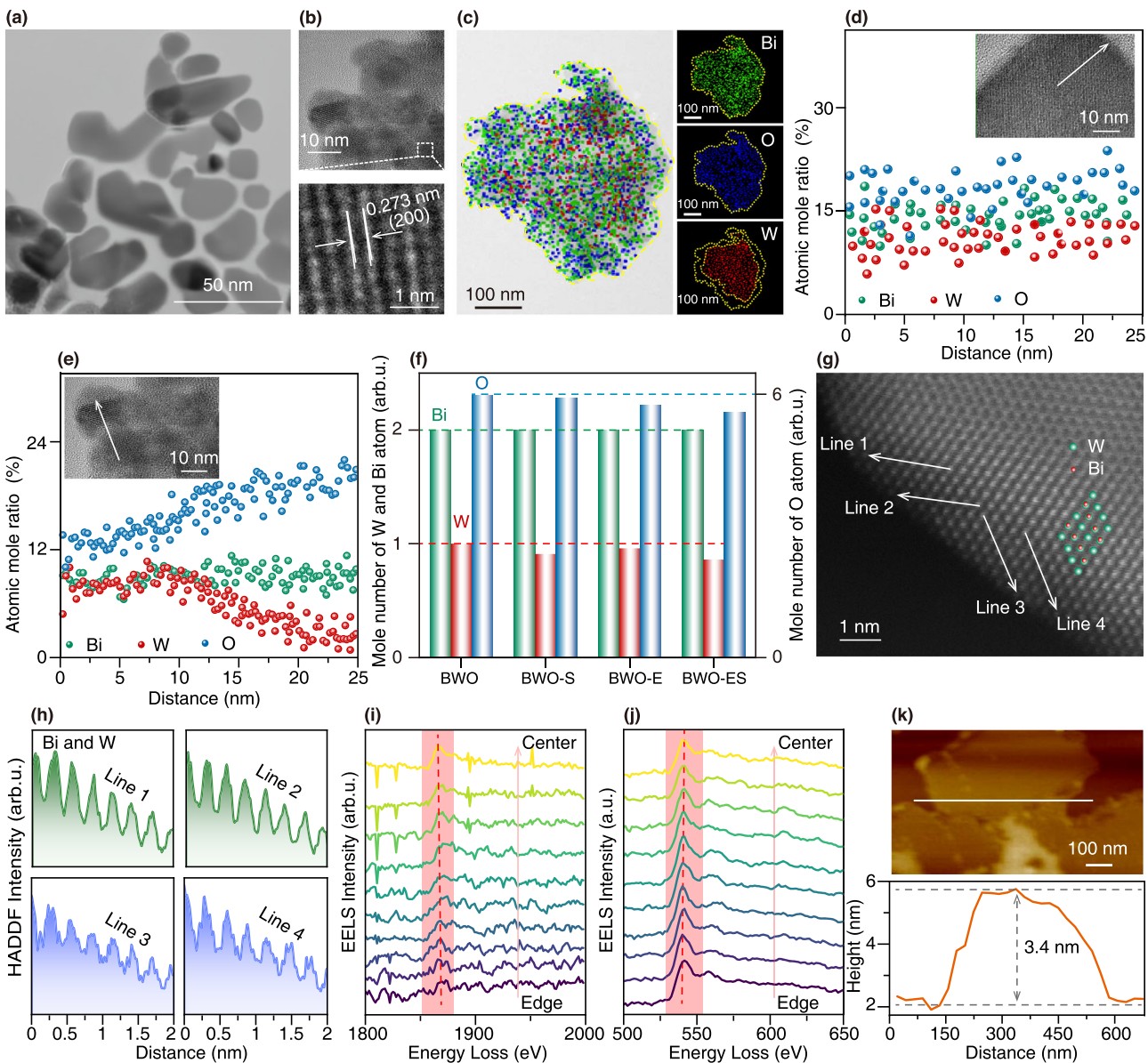

**Fig. 2 | W and O vacancies spatial distribution. a** TEM image, **b** HRTEM image, **c** EDS elemental mappings of the BWO-ES, average values of HRTEM EDS elemental line scanning along the white line in the inset of **d** BWO and **e** BWO-ES, **f** mole number of Bi, W and O atoms of BWO, BWO-S, BWO-E and BWO-ES based on the XRF, **g** atomic-resolution HAADF-STEM image of BWO-ES (the c-axis view of the BWO unit cell), **h** the line profiles along the white lines in **g**, **i** W M4-edge and **j** O K-edge STEM-EELS spectra of BWO-ES (red shaded area and dashed line mark the W M4-edge and O K-edge characteristic peaks in **i** and **j** the red arrow denotes the scanning direction of the STEM-EELS spectra, which progresses from the sample edge toward the center), **k** AFM image and the corresponding cross-section profiles of BWO-ES. (For the results presented in **a**, **b**, **d**, **e**, **g**, and **k**, each measurement was independently repeated five times).

around 3.4 nm, indicative of an exceptionally thin structure (Fig. 2k). In comparison to the BWO, BWO-S, and BWO-E samples, which had thicknesses of 5.9, 6.0, and 4.5 nm, respectively, the BWO-ES exhibited the smallest thickness (Supplementary Fig. 10).

To provide direct evidence of O vacancy incorporation into $Bi_2WO_6$, electron paramagnetic resonance (EPR) spectroscopy was conducted. Notably, BWO-ES exhibits significantly stronger O vacancy defects ($g = 2.0036$) than BWO, BWO-S, and BWO-E, aligning well with the HRTEM EDS elemental line scanning and XRF results (Fig. 3a). Surface chemical states of the associated elements were characterized by X-ray photoelectron spectroscopy (XPS). Defect introduction causes a shift in the W 4f peaks of the BWO-S and BWO-E and BWO-ES to higher binding energies relative to that of the BWO (Fig. 3b). Considering the inverse proportionality between binding energy and

electron density[24], W atom absence causes nearby W atoms to undergo electron rearrangement to compensate for the missing charge through the shared O atoms, resulting the reduced electron density and increase the binding energy of W 4f electrons[18]. In addition, the presence of O vacancies induces changes in electron density and tungsten reduction, leading to a shift of the W 4f binding energy to higher values[25]. The O 1s spectra presented in Fig. 3c demonstrate the distinct binding energies for lattice oxygen (Bi-O and W-O), oxygen vacancies, and surface adsorbed oxygen[5,26,27]. Specifically, BWO exhibits peaks at 529.78, 531.00, and 531.98 eV, whereas BWO-S, BWO-E and BWO-ES display a shift in their binding energy peaks towards lower values. Notably, the BWO-ES sample demonstrates the most substantial shift, with binding energy changes of −0.41, −0.32, and −0.26 eV, respectively. The introduction of W and O dual defects leads to electron

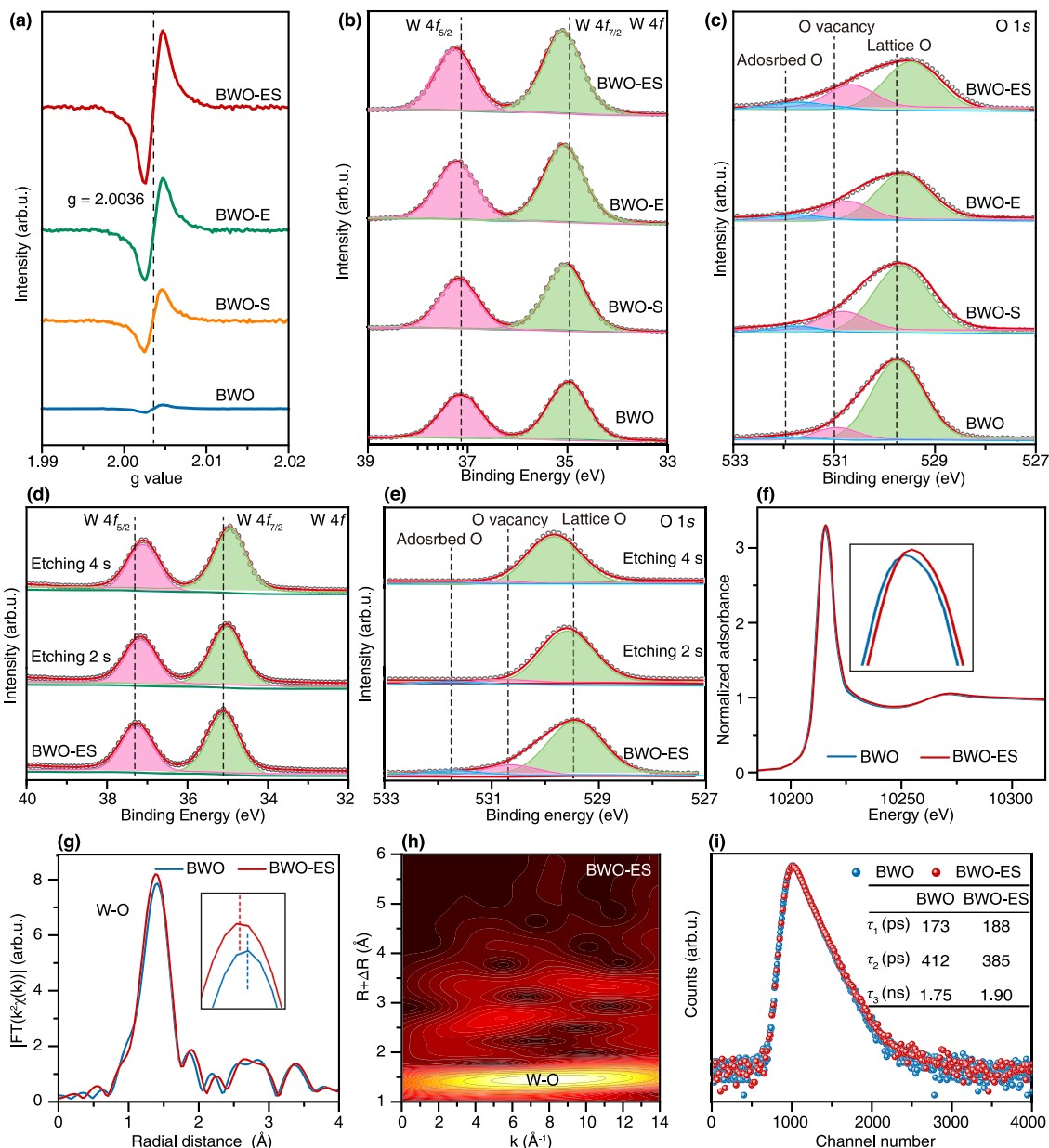

**Fig. 3 | Chemical states characterizations for BWO and BWO-ES. a** EPR spectra (the black dashed line indicates the position in the EPR spectrum where the g value equals 2.0036), high-resolution XPS spectra of the **b** W 4*f* and **c** O 1*s* of BWO, BWO-S, BWO-E and BWO-ES, in-situ Ar plasma etching XPS spectra of **d** W 4*f* and **e** O 1*s* of BWO-ES, **f** W L-edge XANES spectra, **g** W L-edge Fourier-transformed EXAFS k 3χ data of BWO and BWO-ES (the inset provides an enlarged illustration focusing on the region associated with the strongest peak), **h** WT-EXAFS of BWO-ES and **i** positron annihilation lifetime spectra of BWO and BWO-ES ($\tau_1$, $\tau_2$, and $\tau_3$ are three lifetime components, respectively).

redistribution, where the remaining oxygen atoms experience enhanced electron density due to charge compensation effects, thus resulting in lowering the binding energies. Meanwhile, the integrated peak area around 531.00 eV associated with O vacancy exhibits a notably larger value in BWO-S, BWO-E and BWO-ES compared to BWO, suggesting an enhanced concentration of surface oxygen defects in the BWO-S, BWO-E and BWO-ES samples, which is consistent with the results of the EPR and HRTEM EDS elemental line scanning[28]. Furthermore, a shift of approximately 0.32 eV and 0.41 eV towards higher binding energy is observed in the Bi $4f_{5/2}$ and Bi $4f_{7/2}$ peaks of BWO-ES, respectively. This shift provides strong evidence for the formation of higher valence Bi states, which is a direct consequence of the creation of W vacancies (Supplementary Fig. 11)[29].

The depth-dependent distribution of defects throughout the surface and bulk regions was systematically examined using in-situ XPS characterization coupled with controlled argon plasma etching. The invariance of Bi 4*f*, W 4*f*, and O 1*s* XPS spectral features following 2*s* and 4*s* plasma etching procedures indicates uniform chemical environments across both surface and bulk domains of the BWO material (Supplementary Figs. 12, 13 and 14). Depth-profiling XPS analysis revealed the spatial distribution of engineered defects in BWO-ES. After 4*s* of plasma etching, the W 4*f* doublet showed decreased binding

energies (W $4f_{5/2}$: 37.51 to 37.10 eV; W $4f_{7/2}$: 35.21 to 34.91 eV) (Fig. 3d), while the lattice oxygen peak shifted higher (O 1s: 529.49 to 529.84 eV) (Fig. 3e). Notably, the signals corresponding to oxygen vacancies and adsorbed oxygen species were diminished progressively (Fig. 3e), coupled with the Bi $4f$ doublet shift to lower energies (Supplementary Fig. 15). These transitions establish a clear surface-to-bulk defect gradient, confirming the localization of dual-defects within the surface region.

The atomic-level coordination environment of the BWO-ES was characterized through X-ray absorption near-edge structure (XANES) spectroscopy. The W L-edge XANES spectra were collected, characterizing electronic transitions from the $2p$ to $5d$ orbital. The observed higher-energy shift in the W L-edge of BWO-ES, relative to the BWO reference, indicates elevated W valence states, presumably induced by oxygen deficiency in BWO-ES[30] (Fig. 3f). Meanwhile, the enhanced white line peak intensity observed in BWO-ES compared to BWO (Fig. 3f) indicates a higher W oxidation state, which corroborates the XPS-derived oxidation state results. To provide deeper insights into the coordination environment, the Fourier-transformed extended X-ray absorption fine structure spectra and their associated oscillation curves, measured at W L-edges, are also presented in Fig. 3g. Compared to BWO structures, BWO-ES exhibits distinctly shorter W-O distances and more pronounced peak intensities. This structural modification stems from W and O vacancy formation, leading to strengthened interactions between remaining W and O atoms and attributing to increased electron density localization around remaining oxygen atoms following W and O deficiency[18,31]. Furthermore, the dispersion characteristics of W L-edge EXAFS were systematically analyzed using wavelet transformation (WT) across complementary spatial domains ($k$ and $R$ spaces). Spectral analysis revealed a characteristic WT maximum at 8.0 Å$^{-1}$ for BWO-ES (Fig. 3h), corresponding to W-O bonding. Quantitative R-space analysis demonstrated that W and O vacancy incorporation induced a structural modification, reducing the W-O bond length from 1.78 Å in BWO to 1.77 Å in BWO-ES (Supplementary Fig. 16 and Supplementary Table 2)[4,32,33]. To gain insight into the precise dimensions of vacancy-type defects, positron annihilation lifetime spectroscopy (PALS) was performed, as illustrated in Fig. 3i and Supplementary Table 3. The tri-exponential analysis of PALS profiles revealed three lifetime components. The longest lifetime, $\tau_3$, around 1.90 ns, is due to the formation of positronium in large voids. The intermediate lifetime, $\tau_2$, around 385 ps, represents larger volume defects, including oxygen vacancies, and interfaces[34]. The shortest lifetime, $\tau_1$, around 188 ps, is attributable to small vacancies, such as single vacancies and shallow positron traps. Smaller vacancies, such as single vacancies, decrease average electron density, thus extending $\tau_1$. The extended $\tau_1$ value ($\Delta\tau = 15$ ps) in BWO-ES relative to BWO demonstrates the existence of atomic-level vacancies, consistent with reduced electron density distributions[35]. The average positron lifetime (e$^+$) in BWO-ES 1190.9 ps was longer than in BWO 1114.3 ps, indicating that BWO-ES may have a higher concentration or volume of defects. These defects introduce deep-level trap states that can capture photogenerated carriers, inhibit their recombination, and thereby operate as an effective charge reservoir.

The photocatalytic flotation agent degradation (ODA and DMP) efficiency of the synthesized samples was evaluated under visible light irradiation (Text S2 and S3, Supplementary Figs. 17 and 18). As shown in Fig. 4a, b, the BWO-ES sample demonstrated remarkable degradation efficiency, achieving complete degradation of both ODA and DMP within approximately 2 h. In contrast, the pristine BWO showed only approximately 17% degradation efficiency over the same time period. The degradation kinetics were fitted using a zero-order model, consistent with the conditions where the initial pollutant concentration greatly exceeds the adsorption capacity of the photocatalyst surface, leading to surface saturation and rate limitation by intrinsic surface reactions. Through zero-order kinetic fitting (Supplementary Fig. 19),

the degradation rates of BWO-ES for ODA and DMP were determined to be 13.3 and 14.4 mg L$^{-1}$ h$^{-1}$, respectively. These values represent a 3.6-fold enhancement compared to the pristine BWO, and significantly surpass the performance of single-defect BWO-E (6.7 and 7.3 mg L$^{-1}$ h$^{-1}$) and BWO-S (5.0 and 4.3 mg L$^{-1}$ h$^{-1}$). The synergistic effect observed in the BWO-ES photocatalyst, attributed to its in situ fabrication of dual defects, significantly enhances its activity in photocatalytic degradation processes. Subsequently, in the analysis of ODA and DMP degradation products, nitrogen species were closely monitored. The total nitrogen and ammonium (NH$_4^+$) levels remained relatively constant over time, while nitrate (NO$_3^-$) and amino nitrogen (−NH$_2$ − N) exhibited a correlated transformation trend, signifying the complete conversion of −NH$_2$ − N to NO$_3^-$ (Fig. 4c and Supplementary Fig. 20). Meanwhile, the transformation of carbon elements was analyzed through Total Organic Carbon (TOC) detection. After approximately 10 h, the BWO-ES sample achieved complete mineralization of ODA and DMP, confirming the full conversion of carbon elements into CO$_2$ (Supplementary Fig. 21), underscoring the ability of the photocatalytic system to render ODA and DMP entirely harmless.

Tracking the evolution of surface functional groups over time provides a deeper understanding of photocatalytic degradation mechanisms. Through ex-situ Fourier transform infrared spectroscopy, the functional groups of BWO-ES photocatalysts were characterized, as shown in Fig. 4d. After 2 h of dark-state adsorption, distinct characteristic absorption peaks are observed in the infrared spectrum. The asymmetric stretching ($\nu_a$) of the −CH$_3$ end group is observed at 2957 cm$^{-1}$, while the asymmetric ($\nu_a$) and symmetric ($\nu_s$) stretching modes of ·CH$_2$· groups appear at 2925 cm$^{-1}$ and 2854 cm$^{-1}$, respectively[36]. The scissoring mode of $\delta$(−CH$_2$−) and the symmetric deflection of ·CH$_3$ groups ("umbrella mode") are identified at 1467 cm$^{-1}$ and 1389 cm$^{-1}$, respectively[36]. The peak at 1156 cm$^{-1}$ is ascribed to the asymmetric C−O−C stretching vibration ($\nu_a$), a characteristic mode of the morpholine ring[37]. The characteristic absorption peak of the amino group within the morpholine ring was observed at 1624 cm$^{-1}$, which is attributed to the scissoring vibrational mode of the $\delta$(−NH$_2$) group[37]. As the degradation progressed to 2 h, the intensity of all characteristic infrared peaks on the BWO-ES surface decreased to varying degrees, indicating the degradation of nitrogen-containing groups and carbon elements in the DMP molecule. This observation aligns with the results presented in Fig. 4b and Supplementary Fig. 22. By 4 h, the characteristic peak corresponding to the amino group on the morpholine ring had completely disappeared, confirming the complete degradation of nitrogen-containing groups, consistent with the full degradation of BWO-ES observed in Fig. 4b at 2 h. Meanwhile, the characteristic peaks associated with carbon elements in the DMP molecule continued to diminish, nearly vanishing after 12 h, demonstrating complete carbon degradation. The TOC removal efficiency confirmed the complete mineralization of DMP at approximately 10 h (Supplementary Fig. 21). The degradation of ODA followed an analogous trend (Supplementary Fig. 22), revealing that BWO-ES preferentially degrades amino groups in ODA and DMP over carbon elements.

Reactive oxygen species are the key intermediates in photocatalytic degradation, and their identification is vital for understanding the process. Tert butanol (TBA), P-benzoquinone (PBQ), Ammonium oxalate (AO) and Furfuryl alcohol (FA) are the specific scavengers for ·OH, ·O$_2^-$, h$^+$ and $^1$O$_2$, respectively. Within the BWO-ES system for DMP degradation, the introduction of PBQ caused a significant reduction in degradation activity (Fig. 4e). However, the addition of TBA produced no substantial change. Analogously, in the ODA degradation system, PBQ exhibited the highest rate of inhibition, while TBA had virtually no discernible impact (Supplementary Fig. 23). This consistency strongly implies that ·O$_2^-$ is the primary reactive species responsible for the degradation of both ODA and DMP in the BWO-ES system. Furthermore, quantitative assessment of reactive species contributions to

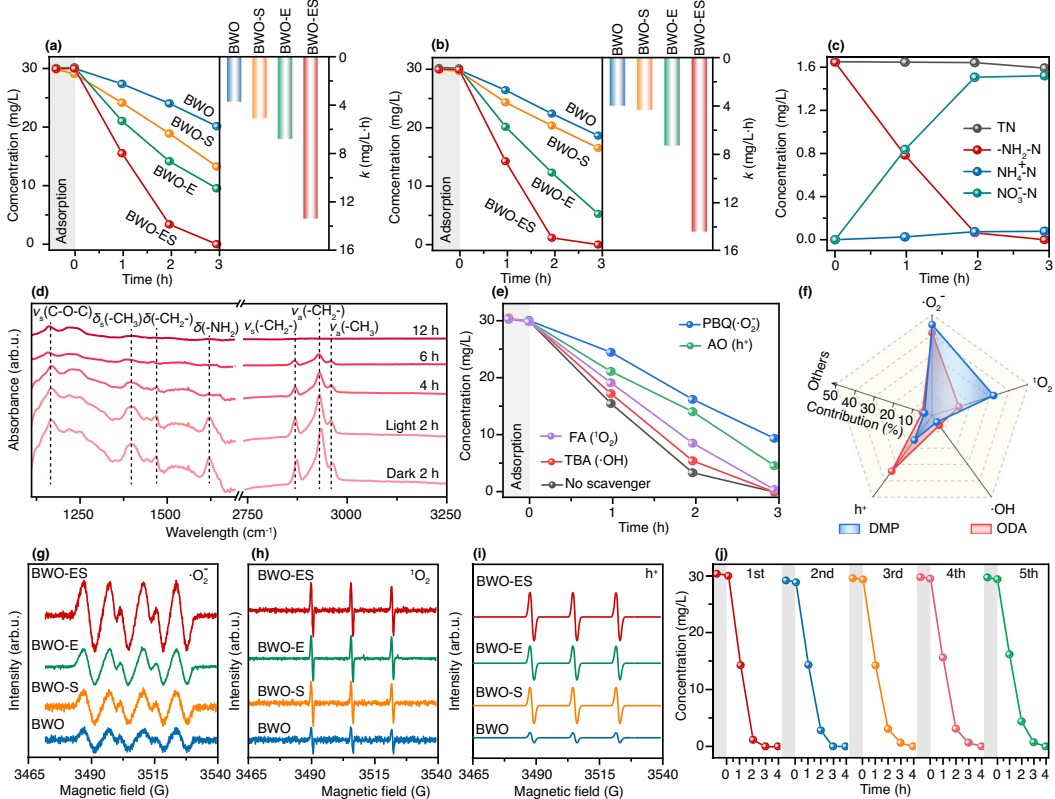

**Fig. 4 | Performance evaluation of BWO, BWO-S, BWO-E and BWO-ES.** Effect of BWO, BWO-S, BWO-E and BWO-ES on the degradation efficiency and reaction rates of **a** ODA and **b** DMP (Reaction conditions: reaction temperature: 25°C, ODA/DMP concentration: 30 mg/L, volume:100 mL, catalyst dosage: 120 mg, initial pH: 3). **c** the temporal evolution of total N, amino nitrogen ($-NH_2-N$), nitrate ($NO_3^--N$), and ammonium ($NH_4^+-N$) during DMP degradation, **d** Ex-situ FTIR spectra of BWO-ES sample with different times. **e** Effect of scavengers on the degradation of DMP using BWO-ES sample (scavenger concentrations: 0.05 mol/L), **f** the contribution of different radicals for the degradation of ODA and DMP in the BWO-ES system. ESR spectra of **g** $\cdot O_2^-$, **h** $^1O_2$, and **i** $h^+$ and **j** recycling stability of BWO-ES sample during the degradation of DMP.

degradation efficiency revealed that $\cdot O_2^-$ was responsible for the predominant proportion of degradation for both ODA (41%) and DMP (45%) (Fig. 4f and Supplementary Fig. 24). DMPO and TEMP-assisted electron paramagnetic resonance (EPR) spectroscopy was employed to detect spin-trapped paramagnetic species, specifically $\cdot OH$, $\cdot O_2^-$, $h^+$ and $^1O_2$[38]. BWO-ES exhibited a markedly stronger reactive species signal intensity than BWO, BWO-S, and BWO-E (Fig. 4g–i). Surprisingly, no EPR signals indicative of $\cdot OH$ were detected in all four photocatalytic systems (Supplementary Fig. 25), highlighting potential differences in reactive species formation mechanisms and aligning with quenching experiment results. The BWO-ES sample demonstrated a remarkable enhancement in the generation of reactive species, with $\cdot O_2^-$, $^1O_2$, and $h^+$ increasing by factors of 8.98, 5.55, and 20.02, respectively, compared to the pristine BWO sample[39,40]. Furthermore, the BWO-ES sample outperformed both BWO-S and BWO-E, demonstrating 3.78-fold, 1.53-fold, and 1.92-fold higher yields than BWO-S, and 2.59-fold, 1.70-fold, and 1.76-fold higher yields than BWO-E for the respective species[39,40]. The substantial improvement is driven by the efficient spatial separation of electrons and holes at the BWO-ES surface, which promotes the generation of reactive species. Consequently, the BWO-ES sample exhibits significantly improved photocatalytic performance, underscoring the critical role of in situ dual defect engineering in enhancing photocatalytic efficiency and advancing material design. Simultaneously, the photocatalytic cycling stability of the BWO-ES sample was further evaluated. After multiple degradation cycles of ODA and DMP, compared to that of the BWO-S and BWO-E (Supplementary Fig. 26), no significant changes in degradation activity or rate were observed (Fig. 4j and Supplementary Fig. 27). This demonstrates that the dual-defect-engineered BWO-ES

sample exhibits excellent stability, highlighting the effectiveness of the dual-defect construction strategy.

In addition, we compared the catalytic degradation of ODA and DMP by BWO-ES with other reported photocatalytic materials (Supplementary Table 4). Some factors in the degradation process, such as catalyst dosage, volume and concentration of organic solution, degradation time, and light source, affect the performance of the photocatalysts. The comparison results revealed that the degradation efficiency of BWO-ES for ODA and DMP was significantly higher than that of the reported photocatalytic materials.

To elucidate the principal factors governing the enhanced efficiency of photocatalytic degradation, the effect of defects on the absorption of solar light was systematically investigated via UV-vis diffuse reflectance spectroscopy (Supplementary Fig. 28a). While W vacancies showed minimal impact, the introduction of O vacancies and W-O dual vacancies markedly enhanced light absorption performance. Bandgap, derived from the Kubelka-Munk function, revealed values of 2.48 eV for BWO-E and 2.54 eV for BWO-ES, significantly lower than the 2.78 eV observed for BWO and BWO-S (Supplementary Fig. 28b). Meanwhile, electronic structure calculations confirmed that O vacancies and W-O dual vacancies reduce the bandgap, whereas the introduction of W vacancies has a negligible effect (Fig. 5a, b and Supplementary Fig. 29). The electronic states are derived from the hybridization of O $2p$, W $5d$, and Bi $6p$ orbitals. Irrespective of defect presence, the predominant transition occurs between the O $2p$ and W $5d$ orbitals. The results demonstrate that O vacancies play a pivotal role in enhancing light absorption properties. The improved absorbance increases the concentration of photogenerated charge carriers, thereby boosting photocatalytic degradation activity.

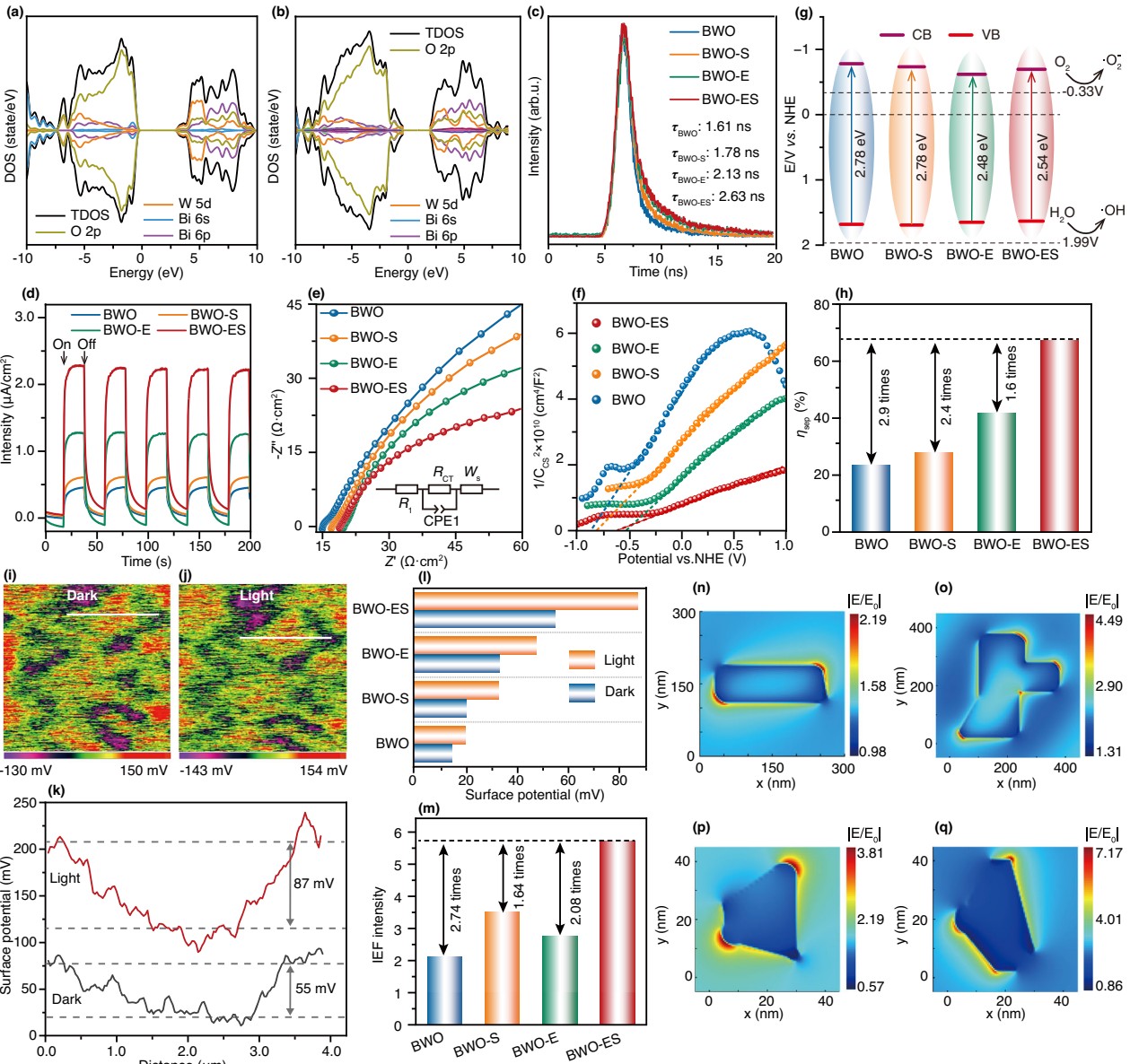

**Fig. 5 | Photo-electrochemical characterizations of BWO, BWO-S, BWO-E and BWO-ES.** Partial density of states of **a** BWO and **b** BWO-ES. **c** time-resolved transient PL spectra of BWO, BWO-S, BWO-E and BWO-ES. **d** transient photocurrent response, **e** Nyquist plots (inset is the equivalent circuit fitting model), **f** Mott-Schottky plots, **g** band diagram schematic, **h** charge separation efficiencies of BWO, BWO-S, BWO-E and BWO-ES. KPFM potential images of BWO-ES **i** in dark and **j** under light, **k** the corresponding line profiles of surface potential along the white line in **i** and **j**. **l** the surface potential and **m** the internal electric field intensity of BWO, BWO-S, BWO-E and BWO-ES. The surface local electric field distribution of **n** BWO, **o** BWO-S, **p** BWO-E and **q** BWO-ES under visible light radiation (wavelength of the incident light above 400 nm).

Therefore, the transport dynamics of photo-generated carriers were further explored through photoluminescence spectroscopy. The BWO-ES sample exhibited the lowest steady-state photoluminescence intensity (Supplementary Fig. 30). In parallel, the time-resolved transient photoluminescence (TRPL) decay spectra revealed that the BWO-ES sample possessed the longest average PL lifetime of 2.63 ns, significantly surpassing the lifetimes of 1.61, 1.78, and 2.13 ns observed in the other samples (Fig. 5c). The rapid transfer of charge carriers is facilitated by the optimized IEF stemming from the co-existence of W and O vacancies[26].

Meanwhile, the surface photovoltage measurements revealed that the BWO-ES sample possessed the highest photovoltage value, suggesting superior charge separation efficiency (Supplementary Fig. 31). This result was further supported by the observation that

BWO-ES also exhibited the largest transient photocurrent response (Fig. 5d). Equivalent circuit fitting of the Nyquist plots indicated that BWO-ES had the lowest charge transfer resistance (61.81 Ω), significantly outperforming BWO (137.5 Ω), BWO-S (98.21 Ω), and BWO-E (88.69 Ω), which highlights its enhanced charge transport properties (Fig. 5e and Supplementary Table 5). Mott-Schottky analysis revealed that the BWO-ES sample exhibits a significantly higher carrier concentration of $1.2 \times 10^{23}\,cm^3$, which markedly exceeds the values observed for BWO ($2.8 \times 10^{22}\,cm^3$), BWO-S ($4.2 \times 10^{22}\,cm^3$), and BWO-E ($5.4 \times 10^{22}\,cm^3$)[5] (Fig. 5f). This effect is likely due to the amplified IEF generated by the surface vacancies in W and O, leading to a substantial increase in the free carrier density. Furthermore, the band structure diagrams of BWO, BWO-S, BWO-E, and BWO-ES were derived from the valence band (VB) XPS results and the band gaps obtained from UV-

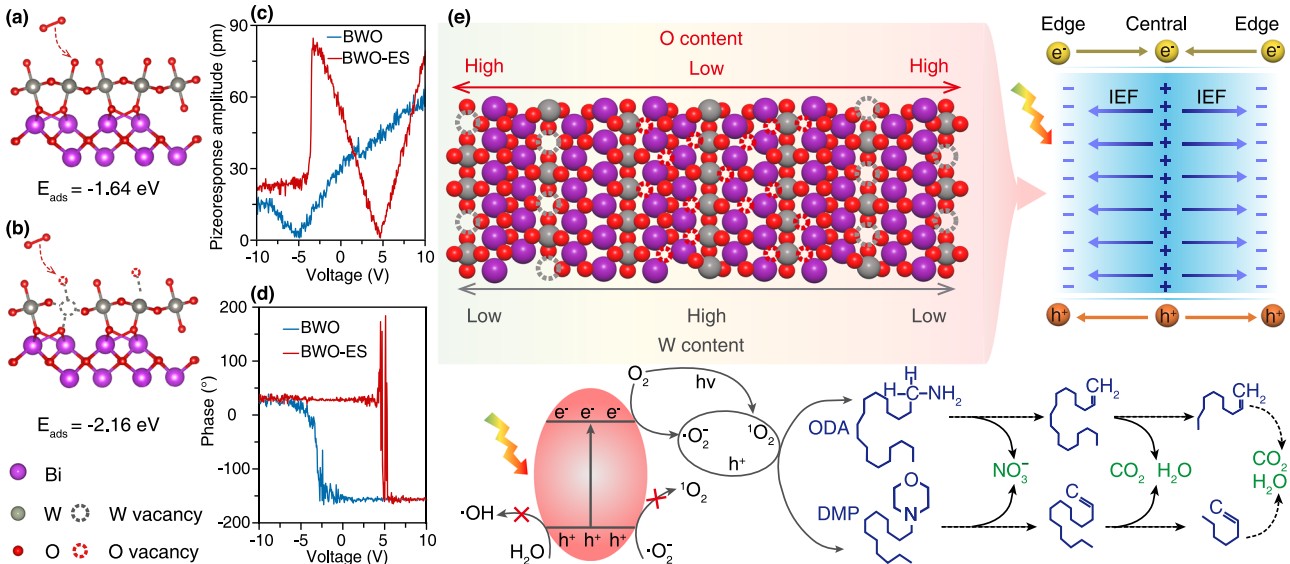

**Fig. 6 | Formation of defect dipole and photocatalytic mechanism.** DFT calculation of $O_2$ adsorbed on **a** O sites in pristine BWO, **b** O vacancies in BWO-ES, **c** piezoresponse amplitude-voltage curves, **d** phase hysteresis loops of BWO and BWO-ES, and **e** the schematic diagram of synergistically generation enhanced dipolar internal electric field through simultaneous incorporation of W and O defects and the photocatalytic degradation mechanisms of ODA and DMP.

visible absorption spectra (Supplementary Fig. 32). The conduction band (CB) positions of all samples were significantly more negative than the formal potential of $O_2/\cdot O_2^-$ (−0.33 V vs normal hydrogen electrode), while the VB positions were below the formal potential of $H_2O/\cdot OH$ (+1.99 V vs normal hydrogen electrode) (Fig. 5g). This demonstrates that $O_2$ in the current system can be efficiently reduced to $\cdot O_2^-$, while the generation of $\cdot OH$ is precluded. These results corroborate the quenching experiments and EPR results discussed previously. Subsequently, a comparative evaluation of photocurrent values, with and without the hole sacrificial agent $Na_2SO_3$, was performed to assess the separation efficiency of surface carriers ($e^-$ and $h^+$) through the equation [41]:

$$\eta_{sep} = \frac{J_{Na_2SO_4}}{J_{Na_2SO_3 + Na_2SO_4}} \times 100\% \qquad (1)$$

Where the $J_{Na_2SO_4}$ and $J_{Na_2SO_3 + Na_2SO_4}$ are the photocurrent response in the presence of the $Na_2SO_4$ electrolyte and mixed solution of $Na_2SO_3$ and $Na_2SO_4$, respectively. The photocurrent density measurements under varying electrolyte conditions revealed that the BWO-ES sample achieved a maximum separation efficiency of 67.5%, significantly outperforming BWO (23.1%), BWO-S (27.7%), and BWO-E (41.7%) (Fig. 5h and Supplementary Fig. 33). By engineering W and O vacancies on the surface of BWO-ES samples, a nanoscale augmentation of the IEF is achieved. This localized intensification enhances the separation efficiency of surface-reaching charge carriers, leading to markedly improved degradation efficiency of ODA and DMP.

To quantitatively evaluate the enhancement of IEF intensity on the material surface through dual defect engineering, Kelvin probe force microscopy (KPFM) was employed to characterize the surface potential under both dark and illuminated conditions. The surface potentials of the BWO-ES sample under dark and illuminated conditions were measured at 55 and 87 mV, respectively (Fig. 5i, j, k). These values significantly exceed those of BWO (15 and 20 mV), BWO-E (34 and 48 mV), and BWO-S (20 and 33 mV) (Fig. 5l and Supplementary Fig. 34). Notably, under illumination, the surface potential of BWO-ES was four times higher than that of the pristine BWO material. Therefore, the internal electric field magnitude was derived from measurements of surface voltage and charge density, in accordance with the model

developed by Kanata-Kito et al.[42–44] (Supplementary Fig. 35). As illustrated in Fig. 5m, BWO-ES demonstrates a substantially enhanced IEF, measuring 2.74, 1.64 and 2.08 times greater than those of BWO, BWO-S and BWO-E, respectively. This pronounced improvement significantly facilitates CST, underscoring its superior ODA and DMP degradation performance. Furthermore, the average sample size, as determined by TEM, was employed in finite element simulations to investigate the optical response of various samples. Under visible light illumination, the BWO-ES sample demonstrated the highest surface local electric field intensity, with edge regions exhibiting significantly greater values than the central zone (Fig. 5n–q). This enhanced electric field reflects stronger light-matter interactions, which are instrumental in boosting photoelectric conversion efficiency. Overall, BWO-ES demonstrates superior photocatalytic performance compared to BWO, BWO-E and BWO-S, exhibiting stronger light absorption capacity, enhanced electron-hole separation, faster interfacial charge transfer, prolonged photogenerated carrier lifetime, and reduced recombination rates. These observations suggest the presence of an IEF within the W and O defect homojunction, which likely drives efficient carrier separation while suppressing recombination.

The $\cdot O_2^-$ is identified as the primary reactive species in the BWO-ES degradation system, formed by transferring photogenerated electrons to the π* orbitals of oxygen molecules[45].

$$O_2 + e^- \rightarrow \cdot O_2^- \qquad (2)$$

Therefore, to elucidate the role of W and O dual defect in $O_2$ adsorption, density functional theory (DFT) calculations were performed (Fig. 6a, b). The $O_2$ adsorption site in BWO is identified as O atoms, whereas in BWO-ES, the adsorption site shifts to O vacancies. The adsorption energy of $O_2$ on BWO is augmented by 0.52 eV when W and O vacancies are introduced in BWO-ES, indicating a stronger charge interaction between $O_2$ and the BWO-ES photocatalyst surface. This enhanced interaction is pivotal in facilitating the $\cdot O_2^-$ generation, thereby boosting the ODA and DMP degradation efficiency. The dual-defect strategy effectively increases the availability of active sites, which serves as a key factor in driving the formation of reactive species during the photocatalytic process.

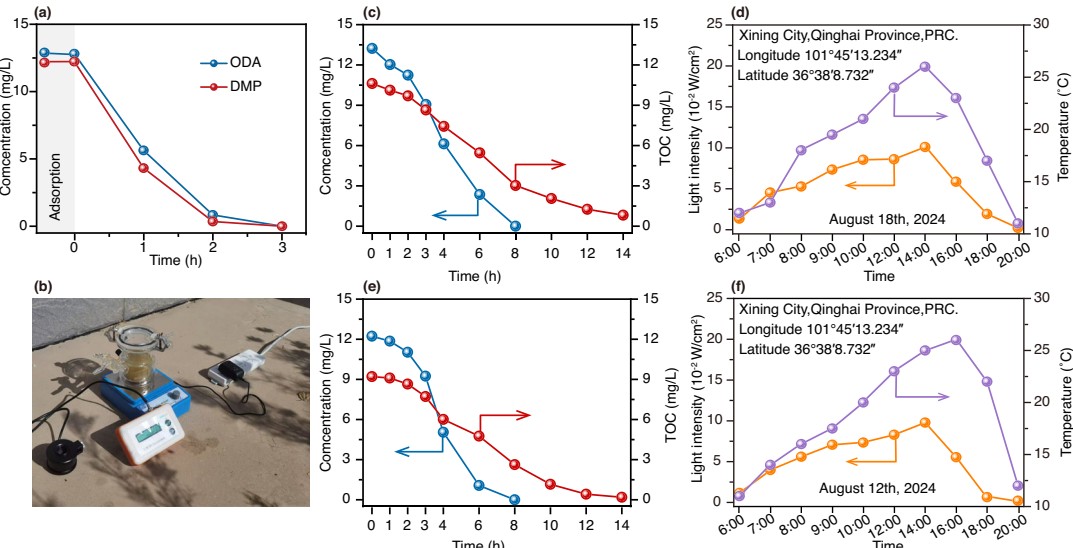

**Fig. 7 | The degradation capability for the real-world flotation tailing wastewater. a** the degradation efficiency of forward and reverse flotation tailing wastewater samples under simulated sunlight conditions using BWO-ES sample (reaction solution: 50 mL, catalyst dosage: 0.06 g). **b** photograph of an outdoor on-site device using solar degradation (reaction solution: 200 mL, catalyst dosage: 0.24 g), the degradation efficiency and TOC removal of **c** the forward and **e** reverse flotation tailing wastewater samples using the BWO-ES sample, variations in temperature and light intensity with time during the degradation process of **d** forward and **f** reverse flotation tailing wastewater samples.

The local ferroelectric switching behavior reveals a profound impact of the engineered defects on the properties of the material. Unlike the pristine BWO sample, which displays a relatively symmetric "butterfly" amplitude loop and a gradual 180° phase reversal centered around zero voltage, the BWO-ES sample exhibits a strikingly asymmetric hysteresis loop with a significant positive voltage shift (Fig. 6c, d). This pronounced imprint effect, where the coercive voltage is shifted away from 0 V, is a hallmark signature of a strong internal bias field that preferentially stabilizes one polarization direction[46]. The origin of this internal field can be attributed to the formation and alignment of defect dipoles, which are composed of associated, oppositely charged tungsten vacancies and oxygen vacancies that were intentionally introduced. Consequently, the alignment of these defect dipoles effectively pins the ferroelectric domain orientation, necessitating a much larger external electric field to induce polarization switching against this built-in field, thereby providing compelling evidence for the existence of defect dipoles and their dominant role in tailoring the electromechanical response of the BWO-ES. In addition, DFT calculations reveal that the z-axis dipole moment of BWO-ES reaches 48.15 D, which is significantly higher than the 1.54 D observed for pristine BWO, thereby providing further confirmation of the presence of defect dipoles (Supplementary Fig. 36). As illustrated in Fig. 6e, the synergistic enhancement of photocatalytic degradation by dual-defect BWO-ES is achieved through the simultaneous introduction of W and O defects on the BWO surface. The formation of this defect dipole is a direct consequence of the spatial separation of charged point defects within the $Bi_2WO_6$ lattice. Specifically, tungsten vacancies act as effective negative charge centers (electron acceptors), while oxygen vacancies behave as localized positive charge centers (electron donors). When these oppositely charged defects are established at distinct lattice positions, they create a net electric dipole moment. This intrinsic, defect-induced electric field is crucial as it can modulate the local electronic band structure and, most importantly, provides an internal driving force for the efficient separation of photogenerated electron-hole pairs, thereby mitigating charge recombination and enhancing overall photocatalytic performance. Additionally, the dual defects increase the number of active sites and strengthen the interaction between $O_2$ and the BWO-ES

surface, facilitating more efficient generation of reactive species and thereby boosting photocatalytic activity. The degradation pathway of ODA and DMP photocatalyzed by BWO-ES was elucidated through GC-MS analysis of intermediate products (Supplementary Data 1 and 2). The process commences with the oxidation of the amino group, followed by the degradation of the carbon chain through an olefin oxidation pathway, resulting in complete mineralization. The toxicity of the intermediate products generally decreases over time, indicating that the photocatalytic degradation process poses minimal ecological and environmental risks (Supplementary Fig. 37).

ODA and DMP are typically found in the flotation tailing wastewater of potash fertilizer plants, which is characterized by its high salinity. As illustrated in Supplementary Table 6, the flotation tailing wastewater sample contains a substantial coexistence of cations and anions, with the concentrations of ODA and DMP being approximately three orders of magnitude lower than those of the ions. It is well established that high salinity-particularly due to elevated chloride ion concentrations-presents major challenges for photocatalytic processes, as $Cl^-$ can act as an efficient hole scavenger, suppressing the generation of reactive oxygen species and thereby reducing photocatalytic efficiency; simultaneously, accumulation of salts can cause photocatalyst deactivation through surface fouling, light blockage, and active site passivation. These factors have traditionally limited the applicability of photocatalytic technologies in saline environments, making the development of photocatalysts that maintain high performance and stability in high-$Cl^-$ matrices a significant achievement in the field. This raises the question of whether the synthesized BWO-ES sample can achieve efficient photocatalytic degradation in actual systems? Initially, a xenon lamp was used to simulate sunlight for the degradation of flotation tailing wastewater samples. As shown in Fig. 7a, complete degradation of ODA and DMP was achieved within approximately 3 h, preliminarily confirming the effectiveness and anti-ion interference capability of the BWO-ES sample in real wastewater samples. Furthermore, outdoor experiments were conducted using sunlight to drive the degradation of flotation tailing wastewater samples, with the setup depicted in Fig. 7b. Under outdoor sunlight conditions, the BWO-ES sample achieved complete degradation of

ODA and DMP in the flotation tailing wastewater samples within approximately 8 h (Fig. 7c and 7e). Additionally, TOC removal results indicated that complete TOC elimination was achieved within around 14 h. Concurrently, ambient temperature and sunlight intensity were monitored throughout the degradation process (Fig. 7d and 7f), providing a comprehensive understanding of the system's performance under real-world conditions. It is demonstrated that the BWO-ES sample, engineered with dual defects, is highly suitable for the solar-light-driven degradation of flotation tailing wastewater, highlighting its practical viability.

## Discussion

In summary, we successfully engineered W and O dual defects in BWO through a one-step in situ defect engineering strategy. Structural and chemical state characterizations confirmed the successful introduction of these defects, with O vacancies predominantly localized in the central region and W defects concentrated at the edges. This unique defect distribution created a nanoscale-enhanced internal electric field on the BWO-ES surface, boosting its photocatalytic performance. Notably, the BWO-ES sample achieved complete degradation of ODA and DMP within 2 h, compared to only 17% for pristine BWO under identical conditions. Specifically, the generation of reactive species ($\cdot O_2^-$, $^1O_2$, and $h^+$) in BWO-ES was enhanced by factors of 8.98, 5.55, and 20.02, respectively, relative to pristine BWO. Photoelectrochemical analyses revealed that BWO-ES exhibits superior light absorption, enhanced electron-hole separation, faster interfacial charge transfer, prolonged carrier lifetimes, and reduced recombination rates. The enhanced photocatalytic activity of BWO-ES is attributed to the localized built-in electric field, which is 2.74 times stronger than that of pristine BWO, alongside a 2.9-fold improvement in charge separation efficiency. Furthermore, BWO-ES demonstrated improved $O_2$ adsorption, facilitating the generation of reactive oxygen species that actively participated in the degradation of ODA and DMP. Notably, BWO-ES exhibited excellent degradation performance for industrial flotation tailing wastewater under solar irradiation, highlighting the practical applicability of the dual-defect engineering strategy. This work paves the way for the rational design of high-performance dual-defect photocatalyst systems, offering new avenues for sustainable environmental remediation.

## Methods
### Chemicals
All chemical reagents used in this work are analytical grade and can be used without further purification (Text S4). All experimental water is deionized water.

### Photocatalyst preparation
Weigh 1.940 g of Bi $(NO_3)_3 \cdot 5H_2O$, 0.658 g of $Na_2WO_4 \cdot 2H_2O$, 0.120 g of cetyltrimethylammonium bromide, Select $Bi_2WO_6$ (60 mL $H_2O$), $Bi_2WO_6$ (60 mL $H_2O$ + 0.08 g NaOH), $Bi_2WO_6$ (60 mL EG), and $Bi_2WO_6$ (60 mL EG + 0.08 g NaOH) prepared as the research objects. Ultrasonic treatment at room temperature for 0.5 h and magnetic stirring for 4 h until the reagents are completely dissolved. Then transfer the mixture to a 100 mL polytetrafluoroethylene reaction kettle and react hydrothermal at 180 °C for 18 h. After cooling, wash several times with anhydrous ethanol and deionized water, and vacuum dry at 60 °C for 10 h. According to the use of sodium hydroxide or ethylene glycol in the preparation process, the samples are labeled as BWO, NaOH-S, BWO-E, and BWO-ES in sequence.

### Characterization
The crystal structure and morphology characteristics of the prepared materials were analyzed and determined by XRD, SEM, TEM, AFM, BET, FT-IR spectroscopy, and Raman spectroscopy. Among them, the atomic arrangement of the material was observed using a high-angle

annular dark field-scanning transmission electron microscope (HAADF-STEM), and the distribution of defect structures in the material was examined in combination with the electron energy loss spectrum (EELS). The specific information is listed in Text S5. The elemental composition, valence state changes, and vacancy information of the samples were characterized by XRF, XPS, EXAFS, PAT, and EPR. In the EXAFS test, a silicon (111) bicrystal monochromator is used to filter the X-ray beam and a metal W foil is used for energy calibration. The testing of positron lifetime spectra was conducted using ORTEC equipment. The specific information is listed in Text S6. The photoelectrochemical characterization of the materials was tested by UV-vis diffuse reflectance spectra, surface photocurrent, surface photovoltage, Mott-Schottky plots, electrochemical impedance spectroscopy, KPFM, piezoresponse force microscopy, steady-state photoluminescence, TRPL spectroscopy and Zeta potential (Text S7). Photoluminescence and time-resolved photoluminescence spectra were recorded on an Edinburgh Instruments FLS1000 spectrofluorometer equipped with a xenon lamp (450 W), 60 W pulse xenon lamps, and a 980 nm diode laser as the excitation sources. The products and active species of the degradation process were tested using EPR and GC-MS, respectively. The changes in TOC were measured using an Analytikyena instrument, and the changes in ion concentration in the solution were investigated using ion chromatography (Text S8).

### Theoretical calculation
The VB, CB, and bandgap width ($E_g$) of different materials were calculated based on DFT using the VASP code (Text S9). The adsorption energies of oxygen molecules on specified crystal planes of BWO and BWO-ES were simulated using the GGA (Generalized Gradient Approximation) method with PBE generalized treatment (Text S10). The local electric field intensity on the surface of the nanostructure was calculated using COMSOL Multiphysics 5.6 software (Text S11). The dipole moments of BWO and BWO-ES were calculated using DFT (Text S12).

## Data availability
All the data supporting the results and findings of this study are provided in the paper and the Supplementary Materials. Source data are provided with this paper. All the raw data relevant to the study are available from the corresponding author upon request. Source data are provided with this paper.

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

## Acknowledgements

This work is supported by the National Natural Science Foundation of China (Grant Nos. 22372021 to J.H., U24A20551 to W.S. and H.L., 12505291 to S.Z.); Qinghai Province Science and Technology Program (International Cooperation Special Project, 2025-HZ-803 to H.L.); Kunlun Talent Program of Qinghai Province to S.Z.; CAS Project for Young Scientists in Basic Research (Grant No. YSBR-039 to H.L.).

## Author contributions

S.Z. conceived and designed the study. L.M. performed the experiments. W.H. and M.D. operated DFT calculations. C.W. and Z.S. provided the discussion. S.Z. wrote the paper. S.Z., J.H., W.S., and X.Y. revised the paper. S.Z., H.L. and X.Y. supervised the experiments. All authors contributed to the analysis.

## Competing interests

The authors declare no competing interests.

## Additional information

[1]Key Laboratory of Green and High-end Utilization of Salt Lake Resources, Qinghai Provincial Key Laboratory of Resources and Chemistry of Salt Lakes, Qinghai Institute of Salt Lakes, Chinese Academy of Sciences, Xining, China. [2]University of Chinese Academy of Sciences, Beijing, China. [3]State Key Laboratory of Fine Chemicals, Frontiers Science Center for Smart Materials Oriented Chemical Engineering, School of Chemical Engineering, Dalian University of Technology, Dalian, China. [4]College of Environmental and Chemical Engineering, Jiangsu University of Science and Technology, Zhenjiang, China. ✉e-mail: siyuanzh@isl.ac.cn; jhou@dlut.edu.cn; swd1978@ujs.edu.cn; yexs@isl.ac.cn

