## [Transparent Peer Review file · Nature Communications]

Region-Specific Defect Engineering of $\text{Bi}_2\text{W}_1-x\text{O}_6-y$ Induces Nanoscale Electric Fields and Surface Active-Sites for Enhanced Visible-Light Oxidation of Salt-Lake Flotation Agents

Corresponding Author: Dr Siyuan Zhang

Version 0:

Reviewer comments:

Reviewer #1

(Remarks to the Author)

In the work, the authors reported a synthesis protocol to achieve W vacancies and O vacancies in the Bi_2WO_6 , and further applied this photocatalyst for ODA and DMP degradation. Overall, the concept of dual vacancies is interesting, but it is lack of solid evidence and moreover, the application of organic pollutant removal is not attractive.

(1) The authors applied EDS heavily on resolving the elements distribution. But the reviewer concerns greatly about the reliability and reproducibility of this method. First, it is the beam spot size in doing this EDS analysis. If the beam spot size is significantly larger your analyzed area, then this data is not very meaningful. Also, this is based on one particles and very localized analysis. More analysis is necessary to verify the above hypothesis. In Figure 2g, the authors have provided a high resolution image, which provides a better choice to analyze Figure 2g with EELS mapping, which can goes to atomic analysis.

(2) From the EXAFS in Figure 3g, it does not show great change in the coordination of the structure. So what is the estimated W and O vacancies concentration ? Also what is their coordination number ?

(3) The authors claimed the presence of W vacancies lead to higher oxidation states as indicated by the XPS data. To the authors, the relationship between W vacancies and high oxidations state is not so strateforward. Can the authors give a detail explanation ?

(4) A longer positron annihilation lifetime does not necessary indicate "effective reservoir for photogenerated carriers".

(5) The performance in Figure 5a and b does not consider the significant surface area difference among different samples. Therefore, the following analysis for the performance difference is less reliable.

(6) The presence of defects does not guarantee the generation of defect dipoles. The KPFM is to show the surface potential change, not the internal electric field. Therefore, more solid evidence is required

Reviewer #2

(Remarks to the Author)

This manuscript presents a pioneering and elegant study on the rational design of a high-performance photocatalyst through region-specific dual-defect engineering. The authors have developed a facile, one-step strategy to create spatially asymmetric W and O vacancies in Bi_2WO_6 nanosheets. The central hypothesis-that localizing W vacancies at the edges and O vacancies in the center creates a "defect dipole" effect to amplify the internal electric field (IEF)-is both highly innovative and convincingly demonstrated. The work represents a significant advance in the field of defect engineering for photocatalysis. It moves beyond the conventional approach of randomly introducing defects and establishes a new paradigm of precise nanoscale spatial control to synergistically enhance charge separation and surface reactivity. The scientific narrative is compelling, flowing logically from material synthesis and characterization to performance evaluation, in-depth mechanistic investigation, and finally, a demonstration of practical application in a challenging, real-world scenario. I am confident it will have a high impact and stimulate further research in the rational design of advanced functional materials. I recommend its publication in Nature Communications after the authors address the following points, which are intended to further strengthen this excellent manuscript.

1. The strategies presently proposed have demonstrated considerable efficacy in constructing anion-cation dual defects within bismuth tungstate materials. Given that Aurivillius phases bismuth vanadate and bismuth molybdate are also layered compounds, could similar defect structures be engineered in these materials as well? Addressing this question would provide compelling evidence for the generalizability of the method.
2. The abstract mentions the "complete oxidation of recalcitrant flotation agents." To be more specific and impactful, I suggest explicitly naming the agents, for example: "...complete oxidation of recalcitrant flotation agents, octadecylamine (ODA) and 4-dodecylmorpholine (DMP), under just 2 h of visible light..." This adds immediate clarity and context for the reader.
3. In the Results section (page 5), the authors attribute the formation of the BWO-ES structure to "controlled solvent selection." While the method is effective, the manuscript would be strengthened by a brief discussion on the underlying chemical reasons. For instance, how do ethylene glycol (EG) and NaOH synergistically promote the formation of W vacancies at the edges and O vacancies in the center?
4. In Fig. 4c, the authors show the conversion of amino nitrogen primarily to nitrate. It would be valuable to comment on the potential formation of nitrite (NO_2^-) as an intermediate. Measuring NO_2^- concentration during the reaction would provide a more complete picture of the nitrogen mineralization pathway. If data is not available, a brief mention of this possibility in the discussion would suffice.
5. In the supplementary information (Fig. S23), the degradation data is fitted to a zero-order kinetic model. While this is a common approximation, photocatalytic reactions often follow pseudo-first-order or Langmuir-Hinshelwood kinetics. A brief sentence in the main text justifying the choice of the zero-order model (e.g., due to the high initial concentration of the pollutant relative to the catalyst's saturation point) would add analytical rigor.
6. The figures are excellent, but some captions could be slightly more descriptive. For example, in the caption for Figure 4, explicitly stating the concentrations of the scavengers used (TBA, PBQ, etc.) would be helpful for reproducibility.
7. The term "defect dipole" is excellent. To strengthen this key concept, add a few sentences in the discussion (Section 3) to elaborate on how the spatially separated, negatively charged W vacancies and positively charged O vacancies create this dipole moment.

Reviewer #3

(Remarks to the Author)

The manuscript NCOMMS-25-44557 exhibits a highly novel and significant advancement in the field of photocatalysis. The concept of "region-specific dual-defect engineering" to create a "defect dipole" effect in Bi_2WO_6 nanosheets is both innovative and elegantly demonstrated. The authors have provided an exceptionally comprehensive and rigorous body of evidence, combining advanced characterization techniques (STEM-EELS, KPFM, PALS), robust performance data, and theoretical calculations (DFT, FDTD) to support their claims. The demonstrated performance, particularly under real-world conditions, is impressive and highlights the practical viability of the engineered material. The manuscript is well-written, logically structured, and the conclusions are strongly supported by the data. It represents a substantial contribution and is highly suitable for publication in Nature Communications after some minor clarifications and improvements.

1. In the Introduction, the authors correctly identify the pollutant problem. To better frame the significance of their results in Figure 7, they could add a sentence explicitly stating the well-known challenges that high salinity (especially high Cl^- concentration) poses to photocatalysis (e.g., hole scavenging, catalyst deactivation), thereby highlighting the exceptional performance of their material.
2. In Figure 2c, the EDS maps are convincing. To further guide the reader, consider adding arrows or circles to the W map to explicitly highlight the areas of reduced W concentration at the edges, as mentioned in the text.
3. For the in-situ Ar plasma etching XPS (Fig. 3d, 3e), please specify the etching parameters (e.g., Ar^+ ion beam energy in keV and current density) in the Methods section. This information is crucial for reproducibility.
4. For the in-situ Ar plasma etching XPS (Fig. 3d, 3e), please specify the etching parameters (e.g., Ar^+ ion beam energy in keV and current density) in the Methods section. This information is crucial for reproducibility.
5. While the double-defect photocatalyst shows excellent stability over five cycles, it is not explicitly stated whether this level of stability is also observed in the materials featuring only single oxygen or tungsten vacancies, as constructed in this study. To clarify this issue and reinforce the validity of the findings, it is advisable to include supplemental cycling data for the single-defect samples.
6. While the adsorption period in the degradation experiments serves as a dark control, a photolysis control (i.e., pollutant solution under illumination without a catalyst) should be explicitly mentioned or shown in the SI to definitively rule out any direct photodegradation of ODA and DMP.
7. The text states complete mineralization was achieved in approx. 10 h (Fig. S20) and 14 h (Fig. 7d, 7f). Please clarify if the difference is due to the different experimental setups (lab vs. outdoor) and initial concentrations. A brief comment would resolve this.
8. The finite element simulations (Fig. 5n-q) were conducted with light from 1050 nm to 1400 nm (NIR range). The photocatalysis experiments were conducted under visible light. This is a potential point of confusion?
9. In the discussion of the real wastewater experiment (Fig. 7), the authors should briefly comment on the potential interfering role of the vast excess of coexisting ions, particularly Cl^- . Highlighting that their catalyst remains highly active despite the presence of these known hole scavengers is a significant point that underscores its robustness and practical advantage.
10. The present dual-defect engineering strategy—characterized by the peripheral distribution of tungsten defects and the central localization of oxygen vacancies—offers a straightforward synthesis approach. However, it remains unclear whether this configurational defect engineering strategy is equally applicable to other layered materials within the Aurivillius phase family. Therefore, it is essential to undertake preliminary experimental studies to validate the universality and effectiveness of this method in alternative Aurivillius phase systems.

Version 1:

Reviewer comments:

Reviewer #1

(Remarks to the Author)

The authors have addressed my concern well and I have no more questions.

Reviewer #2

(Remarks to the Author)

The authors have addressed all comments from reviewers and the manuscript can be accepted for publication in the current version.

Reviewer #3

(Remarks to the Author)

The authors have addressed fully my concernings. Now the revised manuscript can be accepted for publication.

Response to the comments of reviewers

We sincerely appreciate the opportunity provided by the editor and reviewers to revise our manuscript. We are grateful to all reviewers for their meticulous evaluation and insightful comments, which have greatly contributed to improving the quality of our work. In response, we have thoroughly revised both the manuscript and supplementary materials, with all changes clearly highlighted in **red** in the updated versions. Presented below is our detailed, point-by-point response to each reviewer's comment. Reviewer comments are presented in black, while our responses appear in **blue**. All text additions and revisions within the manuscript and supplementary materials are marked in **red** for enhanced clarity. To address the reviewer's issues more effectively, several Tables and Figures that were published or included in the original manuscript and supporting information are referenced in this response and renamed as **Table. R** and **Fig. R**.

Reviewer #1 (Remarks to the Author):

In the work, the authors reported a synthesis protocol to achieve W vacancies and O vacancies in the Bi_2WO_6 , and further applied this photocatalyst for ODA and DMP degradation. Overall, the concept of dual vacancies is interesting, but it is lack of solid evidence and moreover, the application of organic pollutant removal is not attractive.

1. The authors extensively employed EDS to characterize the elemental distribution. But the reviewer concerns greatly about the reliability and reproducibility of this method. First, it is the beam spot size in doing this EDS analysis. If the beam spot size is significantly larger your analyzed area, then this data is not very meaningful. Also, this is based on a single particle and very localized analysis. More analysis is necessary to verify the above hypothesis. In Figure 2g, the authors have provided a high resolution image, which provides a better choice to analyze Figure 2g with EELS mapping, which can goes to atomic analysis.

Response: We thank the reviewer for the thoughtful comments on the reliability and reproducibility of our EDS analysis. To further substantiate the reliability of our EDS results, we conducted six independent elemental line scans on various particles and across different regions of the BWO-ES sample, with the results compiled in **Fig. R1**. These analyses consistently reveal an elemental distribution pattern identical to that presented in **Fig. 2e** of the manuscript. Specifically, a

compositional gradient was observed in every case, characterized by an increasing oxygen concentration and a decreasing tungsten concentration from the center to the edge of the particles. This high degree of reproducibility across multiple measurements confirms the reliability of our findings. Moreover, this analytical approach is well-precedented for characterizing element distribution in published studies on defect photocatalytic materials (*Advanced Materials*, 2023, 35(31): 2302538.). Furthermore, **Fig. R1** was added to the Supplement information as **Fig. S8**.

Fig. R1 Average values of HRTEM EDS elemental line scanning along the white line in the inset

of HRTEM images.

Furthermore, we have conducted extensive experiments using a state-of-the-art HAADF-STEM equipped with a high-efficiency EELS spectrometer. In an ideal scenario, such a technique would provide the most definitive evidence for the local atomic arrangements. While we successfully obtained high-quality, atomically-resolved HAADF STEM images, we found that achieving concomitant EELS maps with true atomic-level resolution was fundamentally precluded by an intrinsic trade-off between the required electron dose and the beam sensitivity of the BWO-ES crystal structure. There is an inherent conflict between achieving atomic spatial resolution and acquiring a sufficient signal-to-noise ratio (SNR) in EELS. The cumulative electron dose needed to generate an atomically-resolved EELS map with an acceptable SNR unfortunately exceeds the critical dose for inducing structural damage in our bismuth tungstate samples. We observed that under the intense, focused electron beam necessary for this type of analysis, the BWO-ES lattice began to amorphize and exhibit local compositional changes before a complete, high-quality map could be acquired. This electron beam-induced damage irretrievably compromises the integrity of the sample, rendering the resulting map unrepresentative of the pristine material. Fortunately, the W M4-edge and O K-edge STEM-EELS spectra of BWO-ES from edge to center (see **Fig. 2i** and **2j** in the manuscript) exhibit the same distribution of W and O elements as the HRTEM EDS elemental line scanning results.

2. From the EXAFS in Figure 3g, it does not show great change in the coordination of the structure.

So what is the estimated W and O vacancies concentration? Also what is their coordination number?

Response: The introduction of dual vacancies in the Bi_2WO_6 lattice resulted in a measurable evolution of its local atomic structure, a key factor governing material properties. The coordination number is a fundamental descriptor of such structural arrangements, and its variation often signals significant changes in bonding environments. Our analysis (Table S2 in Supplementary information) revealed a distinct decrease in the average coordination number for W and O atoms, from 3.32 in the pristine Bi_2WO_6 material to 3.18 in the dual-deficient BWO-ES sample. To corroborate that this structural modification arises from atomic depletion, we employed X-ray fluorescence (XRF) analysis to precisely quantify the elemental composition. The XRF data (**Fig. 2f** in manuscript) confirmed a non-stoichiometric composition, yielding formulas of $\text{Bi}_2\text{W}_{0.92}\text{O}_{5.88}$ (BWO-S),

$\text{Bi}_2\text{W}_{0.94}\text{O}_{5.76}$ (BWO-E), and $\text{Bi}_2\text{W}_{0.83}\text{O}_{5.61}$ (BWO-ES), thereby confirming that the observed reduction in coordination is a direct consequence of the induced W and O vacancies.

Fig. R2 Mole number of Bi, W and O atoms of BWO, BWO-S, BWO-E and BWO-ES based on the XRF

Table R1 EXAFS fitting parameters at the W L-edge for W foil, BWO and BWO-ES.

Sample	Path	N	R (Å)	σ^2 ($\times 10^{-3}$ Å ²)	ΔE_0 (eV)	R -factor
W foil	W-W	8	2.72(0.01)	2.95(0.80)	7.14(1.53)	0.009
		6	3.13(0.01)	2.95(1.12)		
BWO	W-O	3.32(0.58)	1.78(0.02)	3.42(1.85)	2.61(2.68)	0.014
BWO-ES	W-O	3.18(0.48)	1.77(0.01)	2.72(1.51)	2.21(2.32)	0.011

3. The authors claimed the presence of W vacancies lead to higher oxidation states as indicated by the XPS data. To the authors, the relationship between W vacancies and high oxidations state is not so straightforward. Can the authors give a detail explanation?

Response: We thank the reviewer for this insightful comment, which addresses a crucial aspect of our material's defect chemistry. The relationship between the creation of tungsten vacancies and the observed increase in the bismuth oxidation state, as evidenced by the Bi 4f XPS peak shift to higher binding energy, is indeed not a simple one-to-one correspondence but is governed by the fundamental principle of charge neutrality within the crystal lattice. The Bi_2WO_6 crystal is composed of $[\text{Bi}_4\text{O}_2]^{2+}$ and $[\text{WO}_6]^{4-}$ layers, where tungsten exists in a stable +6 oxidation state and bismuth is in a +3 state. The removal of a W^{6+} cation from the lattice creates a significant localized charge deficit, equivalent to a net negative charge of -6. To maintain overall charge neutrality, the

crystal lattice must undergo electronic and structural relaxation to compensate for this deficit. This compensation can occur through several synergistic pathways. The primary mechanism we propose involves the oxidation of neighboring cations. Given that bismuth is the only other cation in the structure, the localized negative potential created by the tungsten vacancies induces an electronic redistribution, prompting adjacent Bi^{3+} ions to release electrons and shift to a higher formal oxidation state, such as $\text{Bi}^{(3+\delta)+}$ or even approaching Bi^{5+} . This oxidation of bismuth cations provides positive charge that partially screens and compensates for the negative charge of the tungsten vacancy. This loss of electron density from the Bi centers reduces the shielding of the Bi nucleus, causing its remaining core-level electrons (e.g., Bi 4f) to be more tightly bound, which is precisely what is observed as a shift to higher binding energy in the XPS spectrum. Concurrently, as our work identifies dual W and O vacancies, the formation of oxygen vacancies serves as another critical charge compensation mechanism. The creation of each oxygen vacancy (by removing an O^{2-} anion) leaves behind two electrons and contributes an effective +2 charge to the lattice. Therefore, the substantial -6 charge from a single W vacancy is likely balanced by a combination of forming multiple oxygen vacancies and the oxidation of neighboring bismuth ions. This synergistic compensation is more energetically favorable than relying on a single mechanism. Thus, the observed shift in Bi 4f binding energy is a direct consequence of the W vacancy, where the oxidation of Bi^{3+} is a key part of the complex charge compensation mechanism required to maintain crystallographic stability.

4. A longer positron annihilation lifetime does not necessary indicate "effective reservoir for photogenerated carriers".

Response: We sincerely appreciate the reviewer's critical perspective on the mechanistic implication between positron annihilation lifetime and the material's function as an effective reservoir for photogenerated carriers. We fully agree that positron annihilation lifetime spectroscopy (PALS) itself is a structural probe: a longer positron lifetime quantitatively reflects an increased concentration or size of open-volume defects (such as vacancies, vacancy clusters, or dislocations), which manifest as regions of locally reduced electronic density, but PALS does not directly measure photocarrier trapping or dynamics. Nevertheless, extensive studies in semiconductor defect physics reveal a robust correlative and sometimes causal chain: many open-volume defects mapped by

PALS-especially cation or anion vacancies-serve as deep-level electronic trap states within the bandgap, which are widely recognized as strong centers for trapping photogenerated electrons or holes (*Journal of the American Chemical Society*, 2024, 146(13): 8787-8799, *Journal of Materials Chemistry A*, 2021, 9(42): 23765-23782). The presence and population of these trap states have a critical influence on carrier lifetimes: deep traps can captivate charge carriers, spatially delay recombination, and thus effectively act as a carrier ‘reservoir’ particularly relevant for photocatalytic and photovoltaic applications where slow release of stored carriers enables more efficient interfacial redox reactions, as supported by time-resolved photoluminescence and transient absorption studies (*ACS Applied Materials & Interfaces*, 2019, 11(43): 40860-40867). Given these points, we readily acknowledge that PALS provides indispensable but indirect, structural insight: it highlights the presence and evolution of defect sites that can serve as carrier reservoirs, but the ‘reservoir’ function per se should be inferred only in conjunction with complementary photophysical measurements. We therefore clarify this logical progression in our manuscript and propose a revision for the originally oversimplified statement.

Therefore, we have been added the following sentences in the revised manuscript on Page 11. “The average positron lifetime (e^+) in BWO-ES 1190.9 ps was longer than in BWO 1114.3 ps, indicating that BWO-ES may have a higher concentration or volume of defects. These defects introduce deep-level trap states that can capture photogenerated carriers, inhibit their recombination, and thereby operate as an effective charge reservoir.”

5. The performance in Figure 5a and b does not consider the significant surface area difference among different samples. Therefore, the following analysis for the performance difference is less reliable.

Response: We extend our sincere gratitude to the reviewer for your constructive suggestions. To eliminate the influence of specific surface area differences on the observed photocatalytic degradation activity, we regulated the amounts of surfactant during the synthesis process while maintaining all other experimental conditions constant. This adjustment allowed us to control the specific surface areas of the BWO, BWO-E, and BWO-S samples to closely approximate that of the BWO-ES sample (**Fig. R3**). Photocatalytic degradation activity tests revealed that, under the current experimental conditions, samples with increased specific surface areas (BWO, BWO-E, BWO-S)

exhibited slightly enhanced degradation activity compared to their respective pristine forms (**Fig. R3**). However, the overall degradation trend remained fundamentally unchanged. As a result, the subsequent discussions and analyses presented in this work are reliable and unaffected by specific surface area variations.

Fig. R3 Nitrogen adsorption-desorption isotherms of (a) BWO, (b) BWO-S, (c) BWO-E and (d) BWO-ES under different CTAB dosage conditions. Degradation profiles of prepared Bi₂WO₆ with different specific surface areas for (e) ODA, (f) DMP.

6. The presence of defects does not guarantee the generation of defect dipoles. The KPFM is to show the surface potential change, not the internal electric field. Therefore, more solid evidence is

required.

Response: We thank the reviewer for this insightful comment. We agree with the reviewer that the mere presence of defects is not sufficient to generate a net internal electric field. However, our work focuses on a specifically engineered spatial separation of defects with opposing effective charges. This spatial charge separation is the fundamental prerequisite for the formation of an electric dipole and, on a nanoparticle scale, a directional internal electric field (IEF). Therefore, the local ferroelectric switching dynamics and the influence of engineered defects on the piezoelectric properties of bismuth tungstate were investigated using switching spectroscopy piezoresponse force microscopy (**Fig. R4**). In BWO, the amplitude-voltage curve retains a nearly symmetric butterfly profile with gradual phase reversal, reflecting intrinsic polarization switching. By contrast, BWO-ES shows a markedly distorted amplitude response and a highly asymmetric phase hysteresis, with a sharp 180° switching occurring only at large positive bias ($\sim +5$ V). This significant internal bias effect can be attributed to the presence of tungsten vacancies at the edges and oxygen vacancies in the central regions, which interact to form stable defect dipoles. These defect dipoles align with the spontaneous polarization and introduce localized internal fields, effectively pinning domains and shifting the coercive voltage. Therefore, these PFM results provide compelling nanoscale evidence for the existence of oriented defect dipoles in the BWO-ES sample, which are directly correlated with the engineered vacancy defects. Meanwhile, density functional theory (DFT) calculations reveal that the dipole strength of BWO-ES reaches 48.15 D (z-axis), which is significantly higher than the 1.54 D observed for pristine BWO, thereby providing further confirmation of the presence of defect dipoles (**Fig. R5**).

Fig. R4 (a) piezoresponse amplitude-voltage curves, and (b) phase hysteresis loops of BWO and BWO-ES.

Fig. R5 The calculated dipole moments for BWO and BWO-ES.

Therefore, the PFM data have been incorporated into **Fig. 6** of the revised manuscript, where they are now presented as **Fig. 6c** and **Fig. 6d**, and the relevant discussion has been supplemented on page 19 as follow.

“The local ferroelectric switching behavior reveals a profound impact of the engineered defects on the properties of material. Unlike the pristine BWO sample, which displays a relatively symmetric "butterfly" amplitude loop and a gradual 180° phase reversal centered around zero voltage, the BWO-ES sample exhibits a strikingly asymmetric hysteresis loop with a significant positive voltage shift (**Fig. 6c** and **6d**). This pronounced imprint effect, where the coercive voltage is shifted away from 0 V, is a hallmark signature of a strong internal bias field that preferentially stabilizes one polarization direction⁴⁶. The origin of this internal field can be attributed to the formation and alignment of defect dipoles, which are composed of associated, oppositely charged tungsten vacancies and oxygen vacancies that were intentionally introduced. Consequently, the alignment of these defect dipoles effectively pins the ferroelectric domain orientation, necessitating a much larger external electric field to induce polarization switching against this built-in field, thereby providing compelling evidence for the existence of defect dipoles and their dominant role in tailoring the electromechanical response of the BWO-ES. In addition, DFT calculations reveal that the z-axis dipole moment of BWO-ES reaches 48.15 D, which is significantly higher than the 1.54 D observed for pristine BWO, thereby providing further confirmation of the presence of defect dipoles.”

The calculated dipole moment of BWO and BWO-ES was added in the revised Supplementary information as Fig. S36.

Furthermore, we concur with the reviewer's accurate statement that Kelvin Probe Force Microscopy (KPFM) directly measures the surface potential, or more precisely, the Contact Potential Difference (CPD) between the conductive tip and the sample surface. However, we wish to clarify that a gradient in the surface potential is a direct and widely accepted manifestation of an underlying internal electric field. Based on the information obtained from KPFM, the apparent internal electric field can be calculated by the Kanata model (*Advanced Materials*, 2022, 34, 2200723, *Advanced Energy Materials*, 2023, 13(11): 2203720). The specific equation is shown below (also see in the Supplementary information on Page 39)

$$E = \sqrt{\frac{-2V_s\rho}{\varepsilon\varepsilon_0}} \quad (R1)$$

where E represents the intensity of the internal electric field; V_s stands for the surface potential detected via KPFM; ρ is surface charge density, which will be obtained by the integral value of the photocurrent density; ε is the dielectric constant of Bi_2WO_6 ($\varepsilon = 80 \text{ F m}^{-1}$); ε_0 refers to the vacuum dielectric constant ($8.854 \times 10^{-23} \text{ J} \cdot \text{K}^{-1}$). The calculated internal electric field intensity was shown in **Fig. R6** (also see **Fig. 5m** in manuscript). The internal electric field intensity of BWO-ES sample was greater than that of BWO, BWO-E, and BWO-S samples.

Fig. R6 The internal electric field intensity of BWO, BWO-S, BWO-E and BWO-ES.

Reviewer #2(Remarks to the Author):

This manuscript presents a pioneering and elegant study on the rational design of a high-performance photocatalyst through region-specific dual-defect engineering. The authors have developed a facile, one-step strategy to create spatially asymmetric W and O vacancies in Bi₂WO₆ nanosheets. The central hypothesis-that localizing W vacancies at the edges and O vacancies in the center creates a "defect dipole" effect to amplify the internal electric field (IEF)-is both highly innovative and convincingly demonstrated. The work represents a significant advance in the field of defect engineering for photocatalysis. It moves beyond the conventional approach of randomly introducing defects and establishes a new paradigm of precise nanoscale spatial control to synergistically enhance charge separation and surface reactivity. The scientific narrative is compelling, flowing logically from material synthesis and characterization to performance evaluation, in-depth mechanistic investigation, and finally, a demonstration of practical application in a challenging, real-world scenario. I am confident it will have a high impact and stimulate further research in the rational design of advanced functional materials. I recommend its publication in Nature Communications after the authors address the following points, which are intended to further strengthen this excellent manuscript.

1. The strategies presently proposed have demonstrated considerable efficacy in constructing anion-cation dual defects within bismuth tungstate materials. Given that Aurivillius phases bismuth vanadate and bismuth molybdate are also layered compounds, could similar defect structures be engineered in these materials as well? Addressing this question would provide compelling evidence for the generalizability of the method.

Response: Building on our success with bismuth tungstate, we have extended the dual-defect engineering strategy to other key Aurivillius phase materials, specifically bismuth vanadate and bismuth molybdate. Preliminary investigations have successfully demonstrated that the developed methodology is equally effective for constructing analogous defect architectures in these photocatalysts. This crucial finding strongly suggests that the dual-defect construction strategy possesses broad generalizability for the entire class of bismuth-based Aurivillius phase materials. A comprehensive study to further elucidate these findings is currently underway, and the results will be presented in a forthcoming publication.

2. The abstract mentions the "complete oxidation of recalcitrant flotation agents." To be more specific and impactful, I suggest explicitly naming the agents, for example: "...complete oxidation of recalcitrant flotation agents, octadecylamine (ODA) and 4-dodecylmorpholine (DMP), under just 2 h of visible light..." This adds immediate clarity and context for the reader.

Response: Based on the suggestions provided by the reviewers, we have added the following sentence in the abstract:

"complete oxidation of recalcitrant flotation agents, octadecylamine (ODA) and 4-dodecylmorpholine (DMP), under just 2 h of visible light irradiation-3.60 times faster than pristine counterparts."

3. In the Results section (page 5), the authors attribute the formation of the BWO-ES structure to "controlled solvent selection." While the method is effective, the manuscript would be strengthened by a brief discussion on the underlying chemical reasons. For instance, how do ethylene glycol (EG) and NaOH synergistically promote the formation of W vacancies at the edges and O vacancies in the center?

Response: We sincerely thank the reviewer for this insightful comment and agree that a detailed discussion of the underlying chemical mechanisms will significantly enhance the manuscript. The spatially controlled formation of dual tungsten (W) and oxygen (O) vacancies in our Bi₂WO₆ synthesis is indeed a direct consequence of the synergistic and distinct chemical roles played by ethylene glycol (EG) and NaOH. EG primarily functions as a high-boiling-point solvent and, more critically, as a mild reducing agent under solvothermal conditions. At elevated temperatures, EG can act as an "oxygen scavenger," abstracting lattice oxygen from the nascent Bi₂WO₆ crystal framework to generate oxygen vacancies. This process is widely documented and is charge-compensated by the partial reduction of adjacent W⁶⁺ cations to lower valence states (e.g., W⁵⁺), a mechanism that stabilizes the oxygen defects. Because EG is the bulk solvent medium, this reductive action occurs throughout the crystal growth process, leading to the formation of oxygen vacancies predominantly within the "center" or bulk of the nanostructures. Concurrently, NaOH establishes a highly alkaline environment. Tungsten oxide species are known to be amphoteric, exhibiting increased solubility in strong alkali to form soluble tungstate anions ([WO₄]²⁻). This high pH promotes a controlled chemical etching or "leaching" of W⁶⁺ ions specifically from the most

chemically reactive and sterically accessible sites of the Bi_2WO_6 nanocrystals-namely, the high-energy edges and corners. This selective removal of tungsten results in the formation of surface-localized tungsten vacancies. Therefore, the synergy arises from two orthogonal chemical pathways: EG acts as a bulk reductant targeting the oxygen sublattice, while NaOH acts as a surface-selective etchant targeting the tungsten sublattice. This unique combination allows for the deliberate engineering of spatially decoupled defects- oxygen vacancies in the crystal core and tungsten vacancies at the crystal edges-which is fundamental to the material's enhanced properties.

Therefore, we will incorporate mechanistic discussion into the revised manuscript on page 5.

“Ethylene glycol, as a reductive polyol, promotes bulk O vacancy generation by abstracting lattice oxygen and stabilizing defect structures. Meanwhile, NaOH, via high alkalinity and potential W leaching, largely localizes W vacancies to surface/edge domains.”

4. In Fig. 4c, the authors show the conversion of amino nitrogen primarily to nitrate. It would be valuable to comment on the potential formation of nitrite (NO_2^-) as an intermediate. Measuring NO_2^- concentration during the reaction would provide a more complete picture of the nitrogen mineralization pathway. If data is not available, a brief mention of this possibility in the discussion would suffice.

Response: We further examined the evolution of nitrogen-containing species by measuring nitrite concentration during the degradation. The data, presented in Fig. R1, conclusively show a lack of nitrite accumulation. The most direct explanation for this result is the system's strongly oxidizing character, which would instantly convert any formed nitrite into its higher oxidation state, nitrate.

Fig. R1 The temporal evolution of total N, amino nitrogen ($-\text{NH}_2\text{-N}$) or morpholine nitrogen ($-\text{C}_4\text{H}_8\text{NO-N}$), and ammonium ($\text{NH}_4^+\text{-N}$), nitrite (NO_2^-) and nitrate (NO_3^-) during (a) ODA and (b)

DMP degradation.

5. In the supplementary information (Fig. S23), the degradation data is fitted to a zero-order kinetic model. While this is a common approximation, photocatalytic reactions often follow pseudo-first-order or Langmuir-Hinshelwood kinetics. A brief sentence in the main text justifying the choice of the zero-order model (e.g., due to the high initial concentration of the pollutant relative to the catalyst's saturation point) would add analytical rigor.

Response: We sincerely thank the reviewer for this insightful and constructive comment. Photocatalytic degradation commonly obeys kinetic behaviors ranging from pseudo-first-order to zero-order, depending heavily on reactant concentration and catalyst surface saturation. The widely recognized L-H mechanism captures this transition: at low substrate concentrations relative to catalyst active sites, reaction rates often express pseudo-first-order dependence on pollutant concentration; however, at elevated concentrations-when the catalyst surface becomes saturated-the kinetics transition to zero-order, as the rate-limiting step shifts from surface adsorption to the inherent surface reaction itself. We have found in multiple experiments that Bi₂WO₆ photocatalyst has almost no adsorption capacity for ODA and DMP, and the degradation behavior is mainly due to the oxidation between active species such as $\cdot\text{OH}$ and $\cdot\text{O}_2^-$ and organic matter. Therefore, the catalyst's active sites were rapidly and comprehensively occupied at the very outset of photoreaction, causing the system to operate within the saturation regime described by the L-H model. In this regime, the reaction is zero-order and independent of substrate concentration. In addition, we attempted to fit the degradation process through a first-order kinetic model, as shown in Fig. R8, where the linear correlation coefficient (R^2) was smaller than that of the zeroth order kinetic model. Therefore, in the article, we fit the degradation process using a zero-order kinetic model.

To improve transparency and address the reviewer's suggestion, we have added the following clarifying sentence in the revised manuscript on page 12.

“The degradation kinetics were fitted using a zero-order model, consistent with the system's operation under conditions where the initial pollutant concentration greatly exceeds the catalyst surface's adsorption capacity, leading to surface saturation and rate limitation by intrinsic surface reactions.”

Fig. R2 Zero-order degradation kinetic model for (a) ODA and (b) DMP, pseudo-first-order degradation kinetic model for (c) ODA and (d) DMP.

6. The figures are excellent, but some captions could be slightly more descriptive. For example, in the caption for Figure 4, explicitly stating the concentrations of the scavengers used (TBA, PBQ, etc.) would be helpful for reproducibility.

Response: The concentration of active species scavengers (TBA, PBQ, etc.) used in the degradation experiment was 0.05 mol/L. Therefore, we have added "scavenger concentrations: 0.05 mol/L" in the caption of Fig. 4e (Page 15).

7. The term "defect dipole" is excellent. To strengthen this key concept, add a few sentences in the discussion (Section 3) to elaborate on how the spatially separated, negatively charged W vacancies and positively charged O vacancies create this dipole moment.

Response: We sincerely thank the reviewer for this positive feedback and for the insightful suggestion to elaborate on the "defect dipole" concept. Therefore, we have updated the discussion in the revised manuscript on Page 19.

"The formation of this defect dipole is a direct consequence of the spatial separation of charged

point defects within the Bi_2WO_6 lattice. Specifically, tungsten vacancies act as effective negative charge centers (electron acceptors), while oxygen vacancies behave as localized positive charge centers (electron donors). When these oppositely charged defects are established at distinct lattice positions, they create a net electric dipole moment. This intrinsic, defect-induced electric field is crucial as it can modulate the local electronic band structure and, most importantly, provides an internal driving force for the efficient separation of photogenerated electron-hole pairs, thereby mitigating charge recombination and enhancing overall photocatalytic performance.”

Reviewer #3(Remarks to the Author):

The manuscript NCOMMS-25-44557 exhibits a highly novel and significant advancement in the field of photocatalysis. The concept of "region-specific dual-defect engineering" to create a "defect dipole" effect in Bi₂WO₆ nanosheets is both innovative and elegantly demonstrated. The authors have provided an exceptionally comprehensive and rigorous body of evidence, combining advanced characterization techniques (STEM-EELS, KPFM, PALS), robust performance data, and theoretical calculations (DFT, FDTD) to support their claims. The demonstrated performance, particularly under real-world conditions, is impressive and highlights the practical viability of the engineered material. The manuscript is well-written, logically structured, and the conclusions are strongly supported by the data. It represents a substantial contribution and is highly suitable for publication.

1. In the Introduction, the authors correctly identify the pollutant problem. To better frame the significance of their results in Figure 7, they could add a sentence explicitly stating the well-known challenges that high salinity (especially high Cl⁻ concentration) poses to photocatalysis (e.g., hole scavenging, catalyst deactivation), thereby highlighting the exceptional performance of their material.

Response: We fully agree that a more explicit statement regarding the challenges posed by high salinity, particularly high chloride (Cl⁻) concentrations, will significantly enhance the contextualization and impact of our results in Fig. 7. High salinity, and specifically elevated concentrations of Cl⁻ ions, constitutes a well-documented impediment to photocatalytic processes aimed at water and wastewater treatment. The primary mechanisms hindering photocatalysis in saline conditions include: (1) competitive hole scavenging by Cl⁻, which markedly suppresses the generation and activity of highly reactive radicals (e.g., ·OH), thereby diminishing pollutant degradation efficiency; and (2) catalyst deactivation or fouling, where the accumulation of salts on the catalyst surface impedes active sites and may also block incident light, further impeding photocatalytic performance. These effects not only hinder pollutant mineralization but can also lead to the formation of undesired and potentially more toxic by-products (e.g., chlorinated organics).

Therefore, we have added a more comprehensive discussion in the revised manuscript on Page 20-21.

“It is well established that high salinity--particularly due to elevated chloride ion concentrations-

presents major challenges for photocatalytic processes, as Cl^- can act as an efficient hole scavenger, suppressing the generation of reactive oxygen species and thereby reducing photocatalytic efficiency; simultaneously, accumulation of salts can cause photocatalyst deactivation through surface fouling, light blockage, and active site passivation. These factors have traditionally limited the applicability of photocatalytic technologies in saline environments, making the development of photocatalysts that maintain high performance and stability in high- Cl^- matrices a significant achievement in the field.”

2. In Figure 2c, the EDS maps are convincing. To further guide the reader, consider adding arrows or circles to the W map to explicitly highlight the areas of reduced W concentration at the edges, as mentioned in the text.

Response: To facilitate a clearer visualization of the actual tungsten distribution boundaries in the EDS mapping presented in Fig. 2c, we have highlighted these regions using yellow dashed circles. This annotation allows for a more intuitive and explicit observation of the diminution of tungsten elements at the periphery.

Accordingly, we have been updated the Fig. 2c in the revised manuscript on Page 7.

Fig. R1 (a) TEM image, (b) HRTEM image, (c) EDS elemental mappings of the BWO-ES, average values of HRTEM EDS elemental line scanning along the white line in the inset of (d) BWO and (e) BWO-ES, (f) mole number of Bi, W and O atoms of BWO, BWO-S, BWO-E and BWO-ES based on the XRF, (g) atomic-resolution HAADF-STEM image of BWO-ES (the c-axis view of the BWO unit cell), (h) the line profiles along the white lines in Figure g, XRD patterns, (i) W M4-edge and (j) O K-edge STEM-EELS spectra of BWO-ES, (k) AFM image and the corresponding cross-section profiles of BWO-ES.

3. For the in-situ Ar plasma etching XPS (Fig. 3d, 3e), please specify the etching parameters (e.g., Ar⁺ ion beam energy in keV and current density) in the Methods section. This information is crucial for reproducibility.

Response: We have added Ar plasma etching parameters in the revised Supplementary information on Page 52.

"The in-situ argon plasma etching system was set to operate with the parameters of 2.0 kV accelerating voltage, beam density was 0.5 mA/cm², and incidence angle was 45°. Under such parameters, the etching rate of the in-situ argon plasma etching system on the Ta₂O₅ standard sample was 1.56 nm/s."

4. For the in-situ Ar plasma etching XPS (Fig. 3d, 3e), please specify the etching parameters (e.g., Ar⁺ ion beam energy in keV and current density) in the Methods section. This information is crucial for reproducibility.

Response: We have added Ar plasma etching parameters in the revised Supplementary information on Page 53.

"The in-situ argon plasma etching system was set to operate with the parameters of 2.0 kV accelerating voltage, beam density was 0.5 mA/cm², and incidence angle was 45°. Under such parameters, the etching rate of the in-situ argon plasma etching system on the Ta₂O₅ standard sample was 1.56 nm/s."

5. While the double-defect photocatalyst shows excellent stability over five cycles, it is not explicitly stated whether this level of stability is also observed in the materials featuring only single oxygen or tungsten vacancies, as constructed in this study. To clarify this issue and reinforce the validity of the findings, it is advisable to include supplemental cycling data for the single-defect samples.

Response: We systematically evaluated the cycle stability of isolated oxygen-vacancy and tungsten-vacancy photocatalysts, as presented in Fig. S26. After five photocatalytic cycles, both materials displayed marked decreases in activity for ODA and DMP decomposition. These results not only reveal their propensity for deactivation, but also highlight the outstanding cycle durability demonstrated by the dual-defect catalyst, which is of particular significance for practical applications demanding sustained performance.

Therefore, we have added Fig. R2 in the revised Supplementary (Fig. S26) materials and added the following description in the revised manuscript on Page 14.

"After multiple degradation cycles of ODA and DMP, compared to that of the BWO-S and BWO-E (Fig. S26), no significant changes in degradation activity or rate were observed (Fig. 4j and Fig. S27)."

Fig. R2 Recycling stability of BWO-S and BWO-E samples during the degradation of (a), (c) ODA and (b), (d) DMP, respectively.

6. While the adsorption period in the degradation experiments serves as a dark control, a photolysis control (i.e., pollutant solution under illumination without a catalyst) should be explicitly mentioned or shown in the SI to definitively rule out any direct photodegradation of ODA and DMP.

Response: To elucidate the specific roles of photolysis and photocatalysis, control experiments were performed in which ODA- and DMP-containing solutions underwent direct light exposure without a catalyst. Over a 3-hour period, only minor reductions in ODA (1.7 mg) and DMP (2.5 mg) concentrations were observed (Fig. R3). These results indicate that photolytic processes contribute negligibly to overall removal, in stark contrast to the complete elimination observed during photocatalytic treatment.

Fig. R3 Photolysis degradation of ODA and DMP.

7. The text states complete mineralization was achieved in approx. 10 h (Fig. S20) and 14 h (Fig. 7d, 7f). Please clarify if the difference is due to the different experimental setups (lab vs. outdoor) and initial concentrations. A brief comment would resolve this.

Response: As pointed out by the reviewer, on the day of the natural light degradation ODA experiment (August 18th, 2024), the highest intensity of sunlight in the city (Xining, China) occurred at 14:00 in the afternoon, at 0.101 W/cm². The indoor degradation experiment was conducted under a xenon lamp light source, and the light outlet positioned 10 cm above the liquid surface, with a light intensity of 0.99 W/cm², which was nearly 9 times that of natural light intensity. After reaching the highest value, the natural light intensity gradually decreased. Therefore, the degradation efficiency under xenon lamp light source irradiation conditions is higher than that of natural light degradation. In addition, the concentration of ODA and DMP solutions prepared during the experiment was 30 mg/L, while the actual concentration of ODA and DMP in brine is generally around 10 mg/L. Therefore, we believe that the differences in experimental conditions and pollutant concentrations are the main reasons for the variation in degradation rates.

8. The finite element simulations (Fig. 5n-q) were conducted with light from 1050 nm to 1400 nm (NIR range). The photocatalysis experiments were conducted under visible light. This is a potential point of confusion?

Response: We sincerely apologize for the inadvertent omission and have meticulously amended this aspect in the revised manuscript on Page 16.

"The local electric field distribution of (n) BWO, (o) BWO-S, (p) BWO-E and (q) BWO-ES under

visible light radiation (wavelength of the incident light above 400 nm)."

9. *In the discussion of the real wastewater experiment (Fig. 7), the authors should briefly comment on the potential interfering role of the vast excess of coexisting ions, particularly Cl⁻. Highlighting that their catalyst remains highly active despite the presence of these known hole scavengers is a significant point that underscores its robustness and practical advantage.*

Response: We fully agree that a more explicit statement regarding the challenges posed by high salinity, particularly high chloride (Cl⁻) concentrations, will significantly enhance the contextualization and impact of our results in Fig. 7. High salinity, and specifically elevated concentrations of Cl⁻ ions, constitutes a well-documented impediment to photocatalytic processes aimed at water and wastewater treatment. The primary mechanisms hindering photocatalysis in saline conditions include: (1) competitive hole scavenging by Cl⁻, which markedly suppresses the generation and activity of highly reactive radicals (e.g., ·OH), thereby diminishing pollutant degradation efficiency; and (2) catalyst deactivation or fouling, where the accumulation of salts on the catalyst surface impedes active sites and may also block incident light, further impeding photocatalytic performance. These effects not only hinder pollutant mineralization but can also lead to the formation of undesired and potentially more toxic by-products (e.g., chlorinated organics).

Therefore, we have added a more comprehensive discussion in the revised manuscript on Page 20.

"It is well established that high salinity--particularly due to elevated chloride ion concentrations--presents major challenges for photocatalytic processes, as Cl⁻ can act as an efficient hole scavenger, suppressing the generation of reactive oxygen species and thereby reducing photocatalytic efficiency; simultaneously, accumulation of salts can cause photocatalyst deactivation through surface fouling, light blockage, and active site passivation. These factors have traditionally limited the applicability of photocatalytic technologies in saline environments, making the development of photocatalysts that maintain high performance and stability in high-Cl⁻ matrices a significant achievement in the field."

10. *The present dual-defect engineering strategy—characterized by the peripheral distribution of tungsten defects and the central localization of oxygen vacancies—offers a straightforward synthesis approach. However, it remains unclear whether this configurational defect engineering*

strategy is equally applicable to other layered materials within the Aurivillius phase family. Therefore, it is essential to undertake preliminary experimental studies to validate the universality and effectiveness of this method in alternative Aurivillius phase systems.

Response: Building on our success with bismuth tungstate, we have extended the dual-defect engineering strategy to other key Aurivillius phase materials, specifically bismuth vanadate and bismuth molybdate. Preliminary investigations have successfully demonstrated that the developed methodology is equally effective for constructing analogous defect architectures in these photocatalysts. This crucial finding strongly suggests that the dual-defect construction strategy possesses broad generalizability for the entire class of bismuth-based Aurivillius phase materials. A comprehensive study to further elucidate these findings is currently underway, and the results will be presented in a forthcoming publication.